# Cysteine availability tunes ubiquitin signaling via inverse stability of LRRC58 E3 ligase and its substrate CDO1

Gisele A. Andree [1,6], Luca J. Stier[1,2,6], Kerstin Schmiederer[1,2], Alina S. Thielen [1,2], Luis Schmid[1], Samuel A. Maiwald [1,2], Karthik V. Gottemukkala[1,2], Jiale Du[1], Susanne von Gronau[1], Claudia Strasser[1], Judith Müller[1], Lukas T. Henneberg[1,3], Camille Guyot[4], Gary Kleiger [5], Matthias Mann [3], Peter J. Murray [4] & Brenda A. Schulman [1,2] ✉

Cellular responses to amino acid fluctuations often hinge on ubiquitin-mediated control of metabolic enzymes, yet the underlying E3 ligase pathways remain poorly defined. Using quantitative proteomics and active cullin-RING ligase (CRL) profiling, we identify LRRC58 as a cysteine-responsive substrate receptor whose stability increases sharply under cysteine starvation. Proteomics reveals an inverse relationship between LRRC58 and the metabolic enzyme cysteine dioxygenase 1 (CDO1), suggesting a cysteine-linked regulatory axis. Biochemical reconstitution and cryo-EM structures show that LRRC58 forms an active CUL2- or CUL5-based CRL that selectively positions CDO1 for ubiquitylation at Lys8. Disease mutant versions of CDO1 mapping to the LRRC58 interface and impaired for the endogenous ubiquitylation pathway were degraded through orthogonal targeting by a VHL-based degrader. Together, our proteomics-guided discovery pipeline, cellular stability studies, and structural analyses uncover a metabolically-tuned LRRC58-CDO1 pathway that links cysteine availability to selective proteasomal turnover, reveals principles of metabolite-regulated CRL activity, and showcases mechanisms distinguishing endogenous and targeted protein degradation.

Regulation of metabolic enzyme levels is critical for maintaining cellular homeostasis in different conditions, such as nutrient limitation or excess, and in response to the availability of essential cofactors and substrates. It has been known for over half a century that enzyme concentrations are not only determined by new protein synthesis, but also by specific degradation pathways responding to changing metabolic cues[1]. In eukaryotes, the ubiquitin-proteasome-system (UPS) is often employed to target specific proteins for degradation[2–4]. Central

to the UPS are E3 ubiquitin ligases, which provide specificity through selectively binding substrates and marking them with ubiquitin signals for degradation.

In human cells, E3 ligases and their cognate substrates have been identified as coordinately responding to metabolic signals including carbohydrates, lipids, ions, metabolic cofactors, and redox stresses[5–14]. Additional modes of regulation have been observed for dipeptides controlling amino acid uptake in yeast and lipid homeostasis in human

[1]Department of Molecular Machines and Signaling, Max Planck Institute of Biochemistry, Martinsried, Germany. [2]Department of Chemistry, School of Natural Sciences, Technical University of Munich, Garching, Germany. [3]Department of Proteomics and Signal Transduction, Max Planck Institute of Biochemistry, Martinsried, Germany. [4]Research Group of Immunoregulation, Max Planck Institute of Biochemistry, Martinsried, Germany. [5]Department of Chemistry and Biochemistry, University of Nevada, Las Vegas, Las Vegas, NV, USA. [6]These authors contributed equally: Gisele A. Andree, Luca J. Stier. ✉e-mail: schulman@biochem.mpg.de

cells[15,16]. Furthermore, some metabolites control the levels of the enzymes catalyzing their biosynthesis and catabolism. An archetypal example of E3 ligase-dependent control of metabolic pathways is the multiprotein budding yeast "GID" E3 ligase complex, discovered in screens for mutants that were "Glucose Induced Degradation Deficient" for a Fructose-1,6-bisphosphatase reporter system[17–22]. The substrate binding subunit of the GID E3 complex is induced by glucose, driving ubiquitin-mediated degradation of its gluconeogenic enzyme substrates under conditions when their activities become superfluous[23,24]. Similarly, when sterol levels are high, they bind SQLE and HMGCR, key enzymes in the cholesterol biosynthetic pathway, to induce their ubiquitylation and degradation[25–34].

Amino acids also affect the stabilities of enzymes that catalyze reactions pertaining to their metabolism[1]. Importantly, cysteine homeostasis has long been known to rely on regulation by the ubiquitin-proteasome pathway. For example, cysteine availability controls the stability of cysteine dioxygenase type 1 (CDO1), which catalyzes oxygenation of cysteine for biosynthesis of hypotaurine and taurine. A series of discoveries in rodents revealed that: (1) dietary cysteine controls CDO1 activity; (2) the cellular abundances of cysteine and CDO1 protein are correlated; (3) limiting cysteine availability triggers degradation of CDO1 by the proteasome; and (4) CDO1 is stable when cysteine is replete[35–37]. Until recently, however, how CDO1 is directed for degradation has been a mystery, at least in part due to a lack of knowledge of its specific E3 ligase(s).

To discover E3 ligases activated and deactivated in response to shifts in environmental conditions, including changes in metabolic state, we developed active cullin-RING ligase (CRL) profiling technology[38]. CRLs are a collection of hundreds of modular E3 complexes, wherein a catalytic cullin-RING module binds a substrate-binding receptor[39,40]. Most CRLs contain either the RING protein RBX1 partnered with CUL1, CUL2, CUL3, or CUL4, or the similar RBX2-CUL5 complex. Each cullin-RING module binds interchangeably to numerous substrate-binding receptors. It is the substrate receptor that specifies the target for ubiquitylation. CRLs are activated by post-translational modification of the cullin subunit with the ubiquitin-like protein NEDD8[41,42]. A neddylated cullin and its partner RING protein recruit and activate a ubiquitin-carrying enzyme (an E2 or ARIH-family RBR E3 ligase) that covalently links ubiquitin to the CRL's receptor-bound substrate[43–47]. Neddylation status of a particular CRL is tightly regulated in accordance with cellular demand for that E3's activity[48–51]. As such, active CRL profiling, which applies quantitative proteomics to affinity-enriched neddylated CRLs, can identify specific substrate receptors whose association with neddylated cullins are modulated in response to changes in metabolic conditions[38].

In this work, to discover E3s regulated by cysteine abundance, we apply active CRL profiling. Assaying multiple cell lines in parallel revealed that cysteine starvation activates a CRL with the substrate receptor, LRRC58. Cellular stability studies revealed that LRRC58 targets CDO1 for degradation in cysteine limiting conditions, and that LRRC58 is destabilized and CDO1 stabilized when its cysteine substrate is abundant. This pathway was independently reported by others using different methodologies while our manuscript was in preparation[52–54]. Our biochemical reconstitution and cryo-EM, in comparison with the targeted protein degradation of CDO1 by a different CRL[55], revealed unique CDO1 ubiquitylation by LRRC58.

## Results

### LRRC58 forms a neddylated CRL upon cysteine starvation
We applied active CRL profiling to lysates from HeLa cells cultured for 24 h in either complete or cysteine-free media, or cultured in complete media for 30, 60, or 120 min following 24 h of cysteine starvation. Immunoprecipitations (IP) performed with our antigen-binding fragment (Fab) that specifically recognizes neddylated CUL1, CUL2, CUL3 and CUL4 (but not other cullins) were analyzed by high-resolution

mass-spectrometry-based proteomics using data-independent acquisition (DIA-MS) (Fig. 1a). When comparing the samples from 24 h cysteine starvation to the control, only one CRL substrate receptor was enriched: the BC-box protein LRRC58 (Fig. 1b). LRRC58 remained enriched after exchange into complete media. Based on these initial experiments, we also examined the active CRLs from HEK293T cells that vary upon a shift in cysteine amounts. Here, the comparison was between cells cultured for 24 h in cysteine-free or in complete media. LRRC58 was the only CRL substrate receptor identified in cysteine starvation samples that was not detected in samples grown in complete media. Furthermore, LRRC58 displayed the highest normalized intensity of all proteins that were detected in cysteine starvation conditions but undetectable when cells were cultured in complete media (Fig. 1c).

Two broad mechanisms have been found to regulate substrate receptor incorporation into a neddylated CRL. CRLs are often regulated through cycles of neddylation and deneddylation, where deneddylated cullin-RING complexes are subject to additional cycles of disassembly and assembly with their repertoire of substrate receptors. In such cases, substrate binding to the receptor inhibits deneddylation and disassembly, shifting the equilibrium of that substrate receptor towards neddylated complexes[48–50]. The abundance of the substrate receptor, often determined by its own ubiquitin-mediated proteolysis, can also impact the extent of its incorporation into an active E3[38,56–58]. As a first step towards uncovering how cysteine regulates LRRC58, DIA-MS experiments were performed to quantify the proteomes of five human cell lines (HEK293T – embryonic kidney, HepG2 – liver tumor, Jurkat – T-ALL, HeLa – cervical cancer, SKBR3 – breast cancer) grown for 24 h in either complete or cysteine-free media. Overall, the presence of LRRC58 protein changes in response to the availability of cysteine in the media, with it primarily being detectable only after culturing in cysteine-free media (Fig. 1d).

### Cellular LRRC58 and CDO1 protein abundance are inversely correlated in a cysteine-dependent manner
To determine if the increased amount of LRRC58 in cysteine starvation conditions is regulated by transcriptional responses, RNA sequencing (RNA-Seq) was performed for HEK293T cells cultured for 24 h in either complete media or cysteine-free media (Fig. 2a). LRRC58 transcript abundances did not significantly differ between the samples, hinting that the response to cysteine starvation may be due to post-transcriptional events. We next examined potential involvement of ubiquitin-mediated degradation by treating cells with either a proteasome or a neddylation inhibitor (MG132 and MLN4924, respectively). In the absence of such inhibitors, LRRC58 was detected by mass spectrometry-based proteomics analysis only in the cysteine starvation condition (notably, we and others[52,54] have been unable to obtain an antibody recognizing LRRC58) (Fig. 2b). Meanwhile, higher amounts of LRRC58 were detected, even in cells cultured in complete media, upon treatment with MG132 and MLN4924 (Fig. 2b). The data raise the possibility that LRRC58 may be continuously degraded under normal growth conditions, but stabilized when cysteine is limiting. Autodegradation is a common mechanism to negatively regulate substrate receptors when their functions are not needed[38,56,57]. In line with this possibility, we could observe efficient LRRC58 auto-ubiquitylation (Supplementary Fig. 1) mediated when in complex with a neddylated cullin-RING scaffold.

We reasoned that proteins with levels that are lower when LRRC58 is expressed (i.e., in the cysteine starvation conditions), but that accumulate in the presence of excess cysteine when LRRC58 abundance is low, could represent putative substrates of this E3. We thus performed DIA-MS to quantify proteins from HEK293T cells cultured in the absence of cysteine, compared with those cultured with a 10-fold excess cysteine concentration. The most abundant protein that increased in the presence of cysteine was CDO1 (Fig. 2c). This

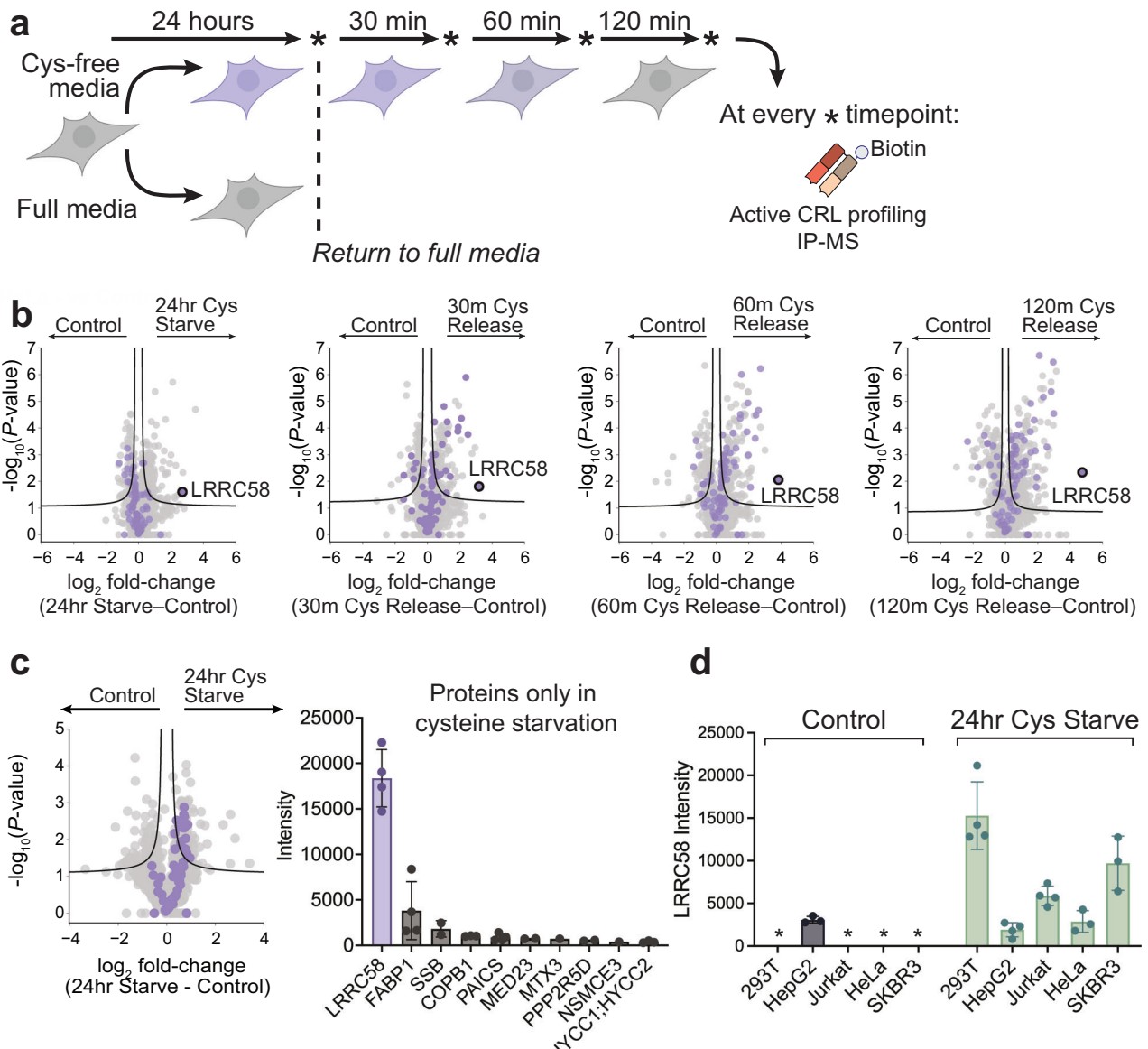

**Fig. 1 | Protein levels increase for the CRL substrate receptor LRRC58 during cysteine starvation. a** Schematic showing the protocol for cysteine starvation of tissue culture cells and time-dependent return to complete media. Substrate receptor levels in active CRLs were assessed through neddylated CRL profiling. Created in BioRender. Andree, G. (2026) https://BioRender.com/zt3humu. **b** Volcano plots highlighting changes in proteins co-immunoprecipitating with neddylated cullins from HeLa cells grown in complete media (control) versus cells grown in cysteine-free media for 24 h prior to switching to complete media at the indicated time points ($n = 3$ independent replicates). Known CRL substrate receptors are colored in purple, all other identified proteins are in gray. Curves for 5% FDR thresholds are shown (two-sided $t$ test, permutation-based FDR calculation,

0.05 FDR, 250 randomizations, s0 = 0.1). **c** Volcano plot (left) same as in (**b**), but with HEK293T cells grown in complete or in cysteine-free media for 24 h ($n = 4$ independent replicates). Bar graph (right) showing the average intensities of the proteins identified in the neddylated CRL IP-MS samples during cysteine starvation that were not detected when cells were grown in complete media. **d** Average LRRC58 intensities from total proteomes of the indicated cell lines grown in either complete (control) or in cysteine-free media for 24 h ($n = 4$ independent replicates). Asterisks indicate samples where LRRC58 could not be detected. All error bars report the standard deviation of the data points. Source data provided as Source Data file.

observation is consistent with the known regulation of CDO1 that depends on cysteine availability[35–37]. Importantly, CDO1 and LRRC58 protein levels were inversely correlated. Peptides corresponding to CDO1 were not detected in cysteine starved samples, whereas LRRC58 abundance was high (Fig. 2d).

An additional four cell lines (HepG2, Jurkat, HeLa, SKBR3, also studied in Fig. 1d) were treated and analyzed similarly. Proteins absent in the cysteine starvation conditions, but detected upon growth with excess cysteine, were considered potential LRRC58 substrates. All such proteins arising from any of the cell lines were modeled for potential interaction with LRRC58 (and its obligate CRL partners Elongin B and

Elongin C, hereafter EloB/C) using HT-Colabfold[59]. Notably, CDO1 scored as a high-potential interactor with the LRRC58-EloB/C complex (Fig. 2e).

To further validate this inverse relationship, we generated knockout (KO) cell lines of LRRC58 and evaluated the CDO1 levels via western blot and DIA-MS. In the wild-type (WT) HEK293T cells, CDO1 was not detected in cysteine starvation conditions. However, in the LRRC58-KO cell line, CDO1 was detectable even in cysteine-starved samples (Fig. 2f, g). Along with genomic verification (Supplementary Fig. 2), DIA-MS analysis of the total proteomes of our LRRC58-KO cell line confirmed the efficacy of the KO (Fig. 2g). Furthermore, we

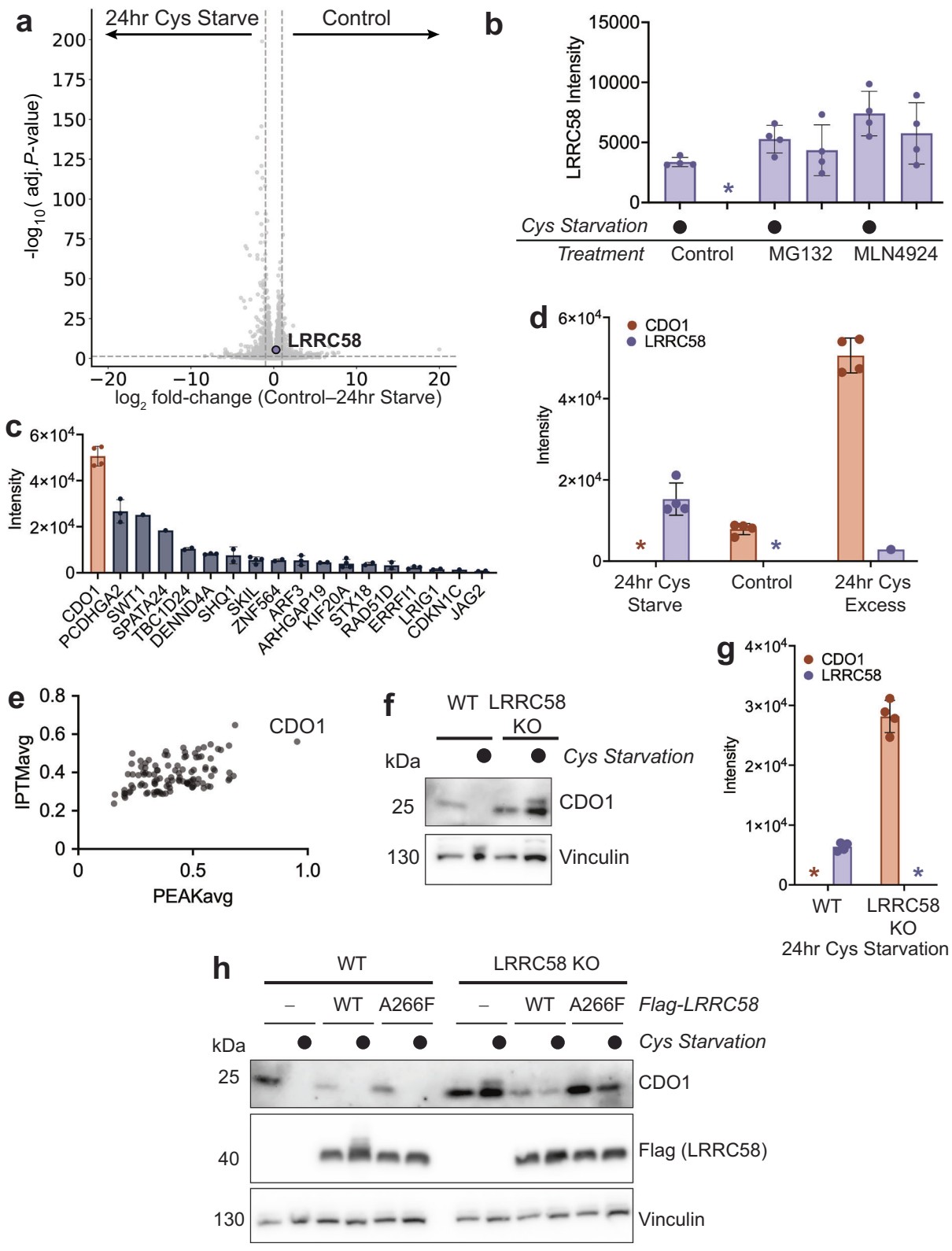

tested whether an overexpression of Flag-LRRC58 (and an A266F mutant control, designed to disrupt EloB/C binding) could restore CDO1 degradation in cysteine starvation conditions in the LRRC58-KO cell line. CDO1 levels indeed decreased upon expression of Flag-LRRC58 WT, whereas expression of Flag-LRRC58 A266F mutant did not result in the same degree of CDO1 loss (Fig. 2h). We also consistently observed the same cysteine-dependent effects on CDO1 level in the WT cell line whether or not Flag-LRRC58 was expressed (Fig. 2h).

## CDO1 stability responds to cysteine starvation in an LRRC58- and CRL2/5-dependent manner

In order to probe the regulation of CDO1 degradation, we used a stability reporter system[60]. CDO1 was expressed as an N-terminal fusion to

**Fig. 2 | Identification of CDO1 as a putative substrate of LRRC58. a** Volcano plot presenting RNA-seq analysis of HEK293T cells cultured for 24 h in complete or cysteine-free media ($n = 3$ independent replicates). Differential expression was assessed using DESeq2[90] by fitting gene-wise negative binomial generalized linear models; statistical significance was evaluated using two-sided Wald tests, with $p$-values adjusted for multiple testing using the Benjamini-Hochberg procedure (adj. $p \leq 0.01$). LRRC58 transcript levels show no significant change. Dashed lines indicate thresholds at 2-fold change and $p$-value = 0.05. **b** Average LRRC58 intensities in HEK293T proteomes ($n = 4$ independent replicates) after 24 h treatment with proteasome (MG132) or neddylation inhibitors (MLN4924) in complete or cysteine-free media. **c** Average intensities of proteins identified in HEK293T proteomes ($n = 4$ independent replicates), which were detectable when cultured in 10-fold excess cysteine but undetectable in cysteine-free media. **d** Average intensities of LRRC58 and CDO1 in HEK293T proteomes after 24 h culture in cysteine-free media, complete media (control), or with 10-fold excess cysteine ($n = 4$ independent replicates). CDO1 was undetectable after culture in cysteine-free media; LRRC58 was undetectable in the control. **e** Average IPTM and PEAK scores from HT-Colabfold analysis of interactions between LRRC58-EloB/C and all proteins absent in cysteine starvation proteomes, but detected with 10-fold excess cysteine for the cell lines; HEK293T, HepG2, Jurkat, HeLa, and SKBR3. **f** Western blot analysis of CDO1 levels in WT-HEK293T and LRRC58 CRISPR-Cas9 knockout (KO) lysates after 24 h culture in cysteine-free media. **g** Average intensities of LRRC58 and CDO1 in WT-HEK293T and LRRC58-KO proteomes after 24 h culture in cysteine-free media ($n = 4$ independent replicates). LRRC58-KO is confirmed by the absence of LRRC58; CDO1 is detectable only in KO. **h** WT-HEK293T and LRRC58-KO cells were transiently transfected with Flag-LRRC58 (WT, or A266F variant (disrupts EloB/C binding)). Western blot of CDO1 levels 24 h post-transfection, followed by 24 h culture in cysteine-free or complete media. All blots are representative of $n = 2$ technical replicates. Flag and vinculin serve as transfection and loading controls, respectively. All error bars report standard deviation. Purple and orange asterisks indicate samples where LRRC58 or CDO1, respectively, were undetectable. Source data provided as Source Data file.

a fluorescent mCherry-tag in a dual-fluorophore reporter construct (GFP–P2A–mCherry-CDO1) (Fig. 3a and Supplementary Fig. 3a). Reporter-expressing cells are GFP-positive, while the fraction of cells lacking mCherry fluorescence reflects CDO1 degradation. To validate this system, we first confirmed that the CDO1 reporter reflects the cysteine-dependent regulation determined for endogenous proteins by DIA-MS. Indeed, the percentage of mCherry-negative cells agrees with the known regulation: low for growth in cysteine, high for growth in the absence of cysteine, reflecting reporter stabilization (Fig. 3a). Second, we confirmed that destabilization of the CDO1 reporter depends on a cullin-RING E3 ligase. The percentage of mCherry-negative cells is low even for cysteine starvation when cells were treated with the neddylation inhibitor (MLN4924). The reporter also showed stabilization upon proteasome inhibition (MG132) (Fig. 3b). Finally, siRNA-mediated knockdown of LRRC58 also stabilized the CDO1 reporter (Fig. 3c and Supplementary Fig. 3b). Taken together, these data support the experimental validity of using a stability reporter as a surrogate readout for the LRRC58-CRL induced degradation of CDO1.

With the validated reporter in hand, we next sought to discover a cullin scaffold contributing to CDO1 degradation. LRRC58 had been annotated as a BC-box protein binding to EloB/C[61]. EloB/C can recruit BC-box substrate receptors to CUL2 or CUL5. Since our current active CRL profiling only enriches for active CUL1-4 complexes, but does not recognize the unique structural arrangement of neddylated CUL5[38,62], our data suggested LRRC58 could associate with CUL2-RBX1 to form an active E3 (Fig. 1b, c). Published analyses of the amino acid sequences of BC-boxes predicted LRRC58 association with CUL2[61]. However, previous high-throughput interactome studies showed LRRC58 binding to CUL5[63–65].

While the siRNA-mediated knockdown of CUL2 or CUL5 protein levels were robust, CDO1 reporter stabilization was observed only upon knockdown of CUL2 (Fig. 3c). Interestingly, only a double knockdown of both CUL2 and CUL5 led to stabilization of endogenous CDO1 (Fig. 3d), hinting at redundant roles for the two different cullin scaffolds. Complementary data were posted on bioRxiv during our manuscript preparation[52]. That study found an effect of knocking down only CUL2 alone, but additionally found that the co-silencing of *CUL5* with *CUL2* led to maximal stabilization of a similar CDO1 reporter.

### CDO1 binds to LRRC58 and is ubiquitylated by activated CRL complexes in vitro

To further define the basis for LRRC58 regulation of CDO1, we reconstituted interactions with purified components (Supplementary Fig. 4). After mixing purified CDO1 with an LRRC58-EloB/C complex, we observed co-migration by size-exclusion chromatography, indicative of a stoichiometric assembly. Similarly, the addition of CUL2-RBX1 led

to the formation of a CRL-substrate complex. A parallel complex was also formed with CUL5-RBX2, confirming the potential for functional usage of either cullin observed endogenously (Fig. 3d). In line with this, both CRLs ubiquitylated CDO1 in an LRRC58-dependent manner in biochemical assays (Fig. 3e).

### Distinct CDO1 targeting by endogenous versus molecular glue-induced degradation machineries

CDO1 was recently identified as the target of molecular glue degraders hijacking the CUL2 substrate receptor VHL[55]. We investigated mechanistic differences for CDO1 ubiquitylation through its endogenous substrate receptor LRRC58, and VHL together with the small molecule "compound-8" (abbreviated "Cmpd8"). Consistent with the literature, the addition of Cmpd8 to cells destabilized the CDO1 reporter (Fig. 4a).

Our finding that LRRC58 can mediate CDO1 degradation through CUL2 allows comparing two E3s that vary only by substrate receptor. We compared the two mechanisms of CDO1 substrate recruitment using our reconstituted in vitro ubiquitylation assays (Supplementary Fig. 5a). These biochemical experiments employed neddylated CUL2-RBX1 that is utilized by both E3s. In vitro ubiquitylation of CDO1 by VHL-CUL2 depended on Cmpd8.

Both the cellular degradation reporter assay and in vitro ubiquitylation assays (Fig. 4 and Supplementary Fig. 5a) showed that CDO1 targeting was more efficient by targeted protein degradation than through LRRC58. This could result from the relatively higher degrader-induced E3-substrate affinity[54,55], as well as variation in catalytic geometries that could also impact the efficiency of ubiquitylation. To experimentally unveil such potential differences[44–47,66,67], we asked if the native degradation mechanism is relatively more constrained by selectivity of lysine targeting. We tested the effects of arginine replacements (which cannot accept ubiquitins) for individual lysines in the CDO1 stability reporter (Fig. 4a). Strikingly, a single K8R substitution was impaired for cysteine-dependent destabilization of CDO1. However, all CDO1 variants were readily destabilized by Cmpd8.

These data raised the possibility that Lys8 is preferentially targeted by the LRRC58 E3. Indeed, in ubiquitylation assays using LRRC58, there was little modification of the K8R mutant compared to WT CDO1. This defect was observed in assays using both WT ubiquitin and a lysine-less version ($K_0$-ubiquitin) that cannot form chains. The data suggest Lys8 modification serves as a basis for chain formation in the LRRC58 pathway (Fig.4b and Supplementary Fig. 5). Notably, the Cmpd8-VHL system retained activity towards the K8R mutant CDO1 (Fig. 4b). Overall, our experimental findings reveal that targeted protein degradation approaches can lead to superior substrate turnover by circumventing limitations imposed by native modes of substrate engagement, and that CDO1 is distinctly presented to CRL catalytic machinery by its endogenous substrate receptor LRRC58.

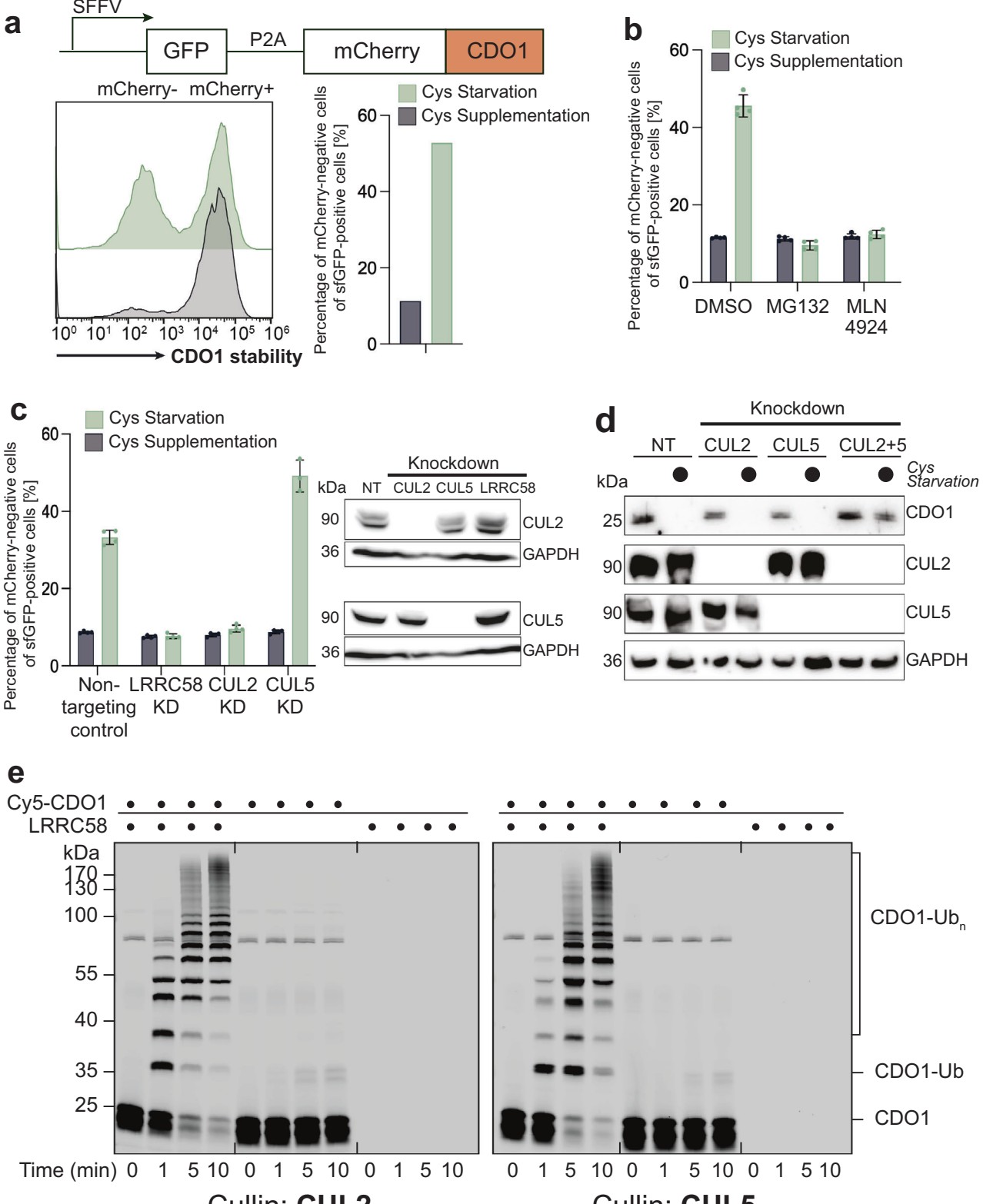

## Cryo-EM reconstructions visualizing CDO1 ubiquitylation by LRRC58 CRLs

To understand the strikingly specific ubiquitylation on CDO1 Lys8 by LRRC58, we sought structural data and performed cryo-EM with various CDO1-LRRC58 CRL complexes. In our initial attempts, we could fit structural models within the density of maps obtained with CUL2 or CUL5, but those maps lacked sufficiently high-resolution features

needed for a mechanistic understanding (Supplementary Figs. 6a, b, 7, 8). Details of catalytic arrangements have been observed in stable mimics of ubiquitylation intermediates obtained by applying our established chemical biology method[44,46,47,68,69]. Our approach simultaneously links the active site of a ubiquitylating enzyme, a modified C-terminus of ubiquitin, and a cysteine substituted for the targeted Lys (here CDO1 Lys8). As a step towards understanding the optimal

**Fig. 3 | Cysteine depletion reduces cellular CDO1 levels in a CRL2/5-dependent manner. a** The CDO1 reporter is a dual-fluorophore reporter construct with GFP, and CDO1 N-terminally tagged with mCherry, separated by a P2A sequence. The CDO1 reporter is monitored by flow cytometry in the presence and absence of extracellular cysteine. A low percentage of mCherry-negative cells (of all GFP-positive cells) reflects CDO1 stabilization, while a higher percentage of mCherry-negative cells (of all GFP-positive cells) indicates CDO1 destabilization, as displayed in the bar graph on the right. **b** Reporter assay showing decreased CDO1 levels in response to cysteine starvation. CDO1 downregulation was sensitive to CRL and 26S proteasome activities (MLN4924 and MG132 treatments, respectively). Bars represent the average values from $n = 4$ independent replicates. **c** Decrease in CDO1 reporter levels upon cysteine starvation is dependent on the presence of LRRC58 and CUL2 but not CUL5. HEK293T cells were treated with siRNAs targeting LRRC58, CUL2, or CUL5 expression or a non-targeting control. Representative Western blot

(right, $n = 2$ independent replicates) showing the efficiency of si-RNA mediated CUL2 and CUL5 knockdown (KD). LRRC58 KD efficiency was determined through analysis of the total proteome (Supplementary Fig. 3b). Bars represent the average values from $n = 4$ independent replicates. All error bars report the standard deviation of the data points. **d** Non-targeting, CUL2, CUL5, or both CUL2 and 5 KD cell lines were grown in either cysteine-free media or complete media for 24 h, then western blots were used to analyze endogenous CDO1 levels and confirm the knockdown efficiency of each cullin. Endogenous CDO1 is only present when cysteine is available, except in the dual CUL2/5 KD ($n = 2$ independent replicates). GAPDH serves as the loading control. **e** In vitro reconstituted ubiquitylation of Cy5-labeled CDO1 in the presence of neddylated CUL2-RBX1 or CUL5-RBX2 and in the absence or presence of LRRC58-EloB/C. Fluorescence scans are representative of $n = 3$ technical replicates. Source data provided as Source Data file.

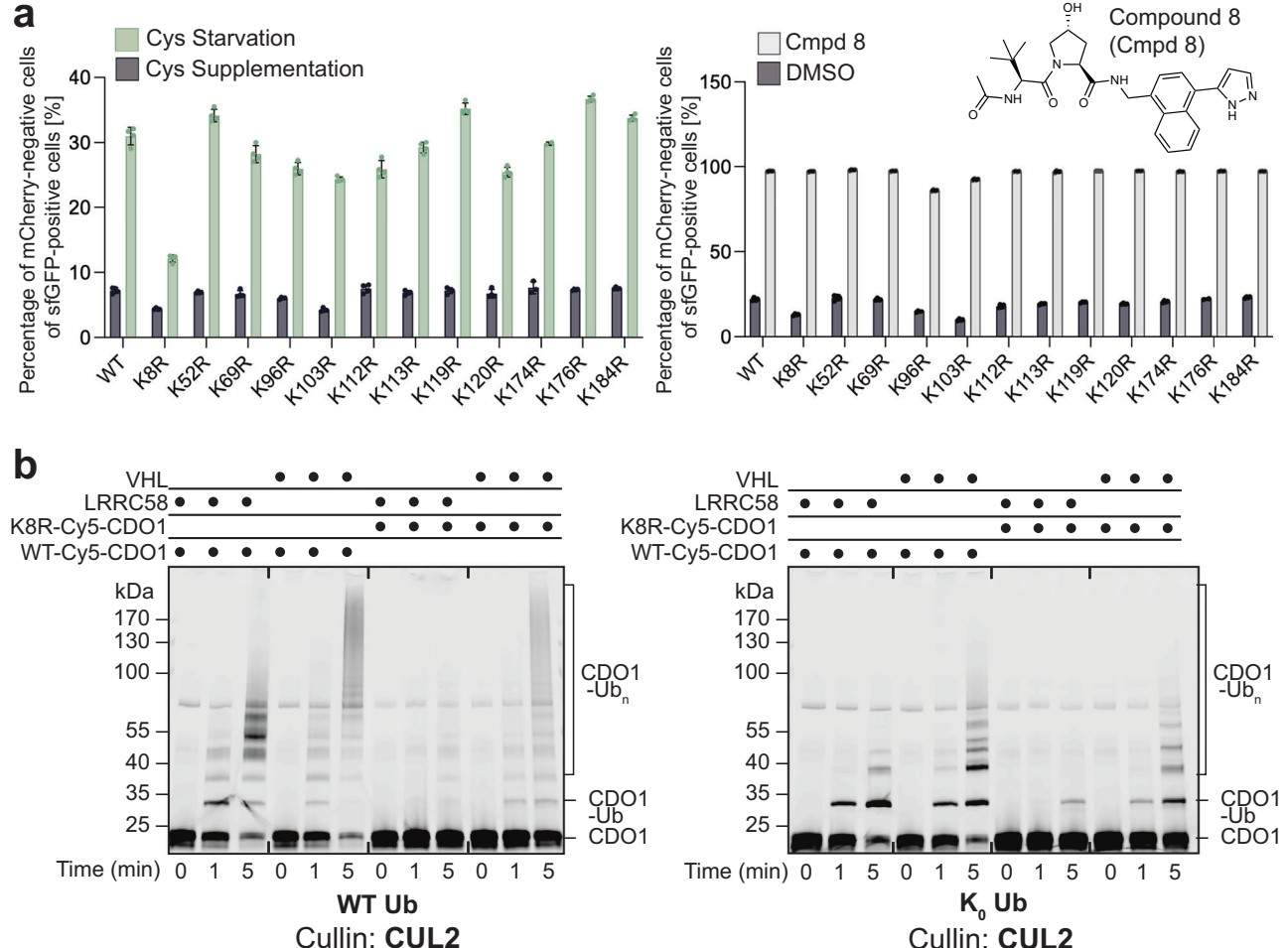

**Fig. 4 | Selective CDO1 Lys targeting during native ubiquitylation is reprogrammed into promiscuous targeting by a molecular glue degrader. a** Left panel displays the reporter assay showing single Lys to Arg CDO1 reporter variants expressed in HEK293T cells grown in complete or cysteine-free media. K8R CDO1 levels were minimally affected upon cysteine starvation in contrast to all other CDO1 mutants, which showed wild-type like responses. The right panel is the same as the left except comparing cells grown in complete media and treated with Compound 8 (Cmpd 8) or DMSO where all CDO1 reporter variants are efficiently degraded. Bars represent the average values from $n = 4$ independent replicates.

Error bars report the standard deviation of the data points. **b** Left panel shows in vitro reconstituted assays comparing Cy5-labeled WT and K8R CDO1 ubiquitylation in the presence of neddylated LRRC58-CUL2 or neddylated VHL-CUL2 with Cmpd8. The efficiency of K8R CDO1 ubiquitylation is lower with LRRC58-CUL2 compared to WT CDO1 but minimally affected in the presence of VHL-CUL2 and Cmpd8. Assay on left was performed with wild-type ubiquitin (WT Ub). The right panel is the same as the left, but performed using a lysine-less ubiquitin ($K_0$-Ub) that cannot form chains. Fluorescence scans are representative of $n = 3$ technical replicates. Source data provided as Source Data file.

reaction components, we examined CDO1 targeting by the different cullins and their various partner ubiquitin carrying enzymes. CDO1 was more rapidly modified by the UBE2L3-ARIH ubiquitylating enzymes (ARIH1 with neddylated CUL2-RBX1, ARIH2 with neddylated CUL5-RBX2), than by UBE2D3 or UBE2R2 (Supplementary Fig. 6c, d). These

results are consistent with prior studies that showed ARIH-family enzymes broadly and efficiently target CRL substrates with a wide range of structural features[43,44,70].

Cryo-EM reconstructions visualized LRRC58 recruitment of CDO1 for ubiquitylation, at 3.73 Å and 2.95 Å overall resolution for

neddylated CUL2-RBX1-ARIH1 and CUL5-RBX2-ARIH2 complexes, respectively (Table 1, Fig. 5a and Supplementary Figs. 9–11). The superior map quality allowed building atomic coordinates for the sample representing CDO1 ubiquitylation by LRRC58 with neddylated CUL5-RBX2-ARIH2 (Supplementary Fig. 12). This, together with previous structures and AlphaFold3 predictions, allowed us to dock sub-complexes to also generate a model representing ubiquitylation by CUL2-RBX1-ARIH1. Despite the lower map quality of the CUL2 complex, these models allow for structural comparisons to be made between the two ubiquitylation machineries employed by LRRC58.

The cryo-EM reconstructions visualize how LRRC58 selectively mediates CDO1 targeting. First, in complexes with CUL2 and CUL5, LRRC58 participates in the canonical CRL architecture (Fig. 5a and Supplementary Fig. 13). Second, LRRC58 makes extensive interactions with CDO1 (Fig. 5a, b). Third, LRRC58 projects CDO1 towards the catalytic portion of a neddylated CRL (Fig. 5a, c). Fourth, ubiquitylation site selectivity results from the constellation of CDO1 residues facing a ubiquitylation active site. Lys8 is the only accessible lysine within 20 Å of the active site in either ubiquitylation assembly (Fig. 5c).

The structure of LRRC58 had not previously been experimentally determined. Our data show the N-terminal portion of LRRC58 forms nine LRRs, which are capped by a unique region at the C-terminus (residues 316–324). This "C-cap" and LRRs4-9 together form the substrate-binding domain (Fig. 5b). The intervening BC- and cullin-box region binds EloB/C and a cullin (Fig. 5d).

Notably, the 3-way interface between EloC, the LRRC58 cullin-box, and cullin were well resolved in the map with CUL5 (Supplementary Fig 12c). Despite the use of this cullin in the complex, comparing with other structures showed LRRC58 displays a canonical CUL2-box, closely resembling that previously determined for the complex between the BC-box protein FEM1B and CUL2 (Supplementary Fig. 14a)[71]. LRRC58's CUL2-box may explain the preferential requirement for CUL2 in determining the stability of the CDO1 reporter. Nonetheless, our data also show a CUL2-box structure is fully compatible with engaging CUL5.

The sequences flanking both sides of LRRC58's cullin-binding region adopted a unique multifaceted and multifunctional structure, centered around a twisted, ~30 Å long two-stranded β-sheet (Fig. 5e). Loops emanating from one end of the sheet form a zinc-bound structure that connects to the cullin-binding region. At the other end of this β-sheet, the loop between both strands serves as a portion of the CDO1-binding site, and as the cap of the LRRs, hence our naming this the C-cap (Fig. 5b, e). Interestingly, additional density could be observed between LRR9, the C-cap, and CDO1, and although its identity remained ambiguous, it might play a role in LRRC58 recruitment of CDO1 (Supplementary Fig. 14b).

## Structural basis for CDO1 ubiquitylation by LRRC58 CRLs

The "top side" of CDO1[72] engages a continuous surface with LRRC58 formed by the six C-terminal LRRs and the C-cap. These interactions primarily map to four patches on CDO1 (Fig. 6a, b). Patch 1 consists of a CDO1 loop displaying Asp168, Gln169, Arg170, and His173. Patch 1 contacts the LRRC58 C-cap centered around Tyr322. Patch 2 is located at the center of the top side of CDO1. Here, Arg141 and Glu143 and the backbone of the adjacent Gly82 contact LRRC58's C-terminal LRR strand (specifically Arg243) and C-terminus. Patch 3 is dominated by the CDO1 loop containing His147, which traverses the LRR5-7 from LRRC58. This His147 side-chain specifically contacts Tyr172 of LRR6. Patch 4 - the CDO1 loop containing Gln55 and Tyr56 - binds the opposite side of the LRRC58 LRR7-9. In addition, we term a fifth CDO1 region (centered around Gln99 and Glu149) the D-patch based on prior studies showing these residues at the heart of the degrader-induced interaction with VHL (Fig. 6b, c)[55]. Notably, Cmpd8-mediated binding of CDO1 to VHL, via the D-patch interactions, presents CDO1 for ubiquitylation in a drastically different orientation than LRRC58 (Fig. 6d)[55].

Roles of the structurally-observed interactions were tested by examining effects of Ala substitutions in each CDO1 patch using our stability reporter system (Fig. 6e–g). In addition, we probed three mutants of CDO1 discovered in patients with rare neurological disorders[73]. Disruption of patch 1 would disrupt the LRRC58 C-cap structure that stabilizes CDO1 binding, most importantly through the pi-stacking interaction of CDO1 His173 with LRRC58 Tyr322 and through CDO1 Arg170 hydrogen bonding to the backbone of LRRC58 Arg323. Mutations in CDO1 Arg141 and Glu143 of patch 2 disrupt contacts with the most C-terminal LRR. The mutation in CDO1 patch 3 abolishes the stabilizing interaction of CDO1 His147 with LRRC58 Tyr172 located in LRR7. The patch 4 mutations of Gln55, Tyr56, Arg57 result in a loss of hydrogen bonding to LRRC58 Glu238. Mutation of patches 1-4 strongly impaired CDO1 instability induced by cysteine-starvation, but not that triggered by the VHL-dependent Cmpd8 (Fig. 6f, g), indicating interactions specific to CDO1 recruitment by LRRC58. In line with the observed defects for residue substitutions at the interface patches, the patient mutants mapping onto patches 2 and 3 (E143K and H147K) show the strongest defects compared to a mutant mapping to a different structural location (A131V, near the CDO1 active site).

Furthermore, during preparation of our manuscript, unbiased saturation mutagenesis in a study posted on bioRxiv independently validated the roles of all four patches on cysteine starvation-induced degradation of similar CDO1 stability reporters[52]. This study also validated the roles for the LRRC58 residues binding CDO1 patches 1–3. In addition, CDO1 patch 3 was also shown to be important for LRRC58-dependent degradation[54].

We uniquely queried the mutants for effects on Cmpd8-induced degradation. These experiments served both as quality control for the stability reporters, and tested specificity. Our data showed these mutational defects were specific for the metabolically-signaled degradation pathway. Conversely, the D-patch variants were refractory to destabilization induced by Cmpd8, but were unstable under cysteine starvation conditions (Fig. 6f, g).

To gain mechanistic insights into the differential targeting of CDO1, we used purified proteins to compare the effects of CDO1 mutations on ubiquitylation by LRRC58 and VHL/Cmpd8 CRLs (Fig. 6e, h, i and Supplementary Fig. 14c). Importantly, the mutational effects on ubiquitylation mediated by the two distinct CRL substrate receptor pathways paralleled those on the two distinct cellular degradation pathways. It is of note that not all mutations completely eliminate ubiquitylation, which implies the importance of the multiple interaction patches.

Finally, the structural models provided insights into the ubiquitylation of a single substrate recruited by a receptor that can employ different cullins. The overall assemblies of the CUL2 and CUL5 complexes are largely similar, evidenced from aligning along the central cullin domains (Supplementary Fig. 15a). However, the structural impact of bending of each cullin is propagated through the elongated assemblies resulting in greater differences at the two ends essential for ubiquitylation: the substrate-receptor assembly at the cullin N-terminus and the catalytic assembly at the cullin C-terminus (Supplementary Fig. 15b, c). Although we cannot rule out that our chemical mimics capture one of a range of conformations along the ubiquitylation trajectories, they nonetheless show how Lys8 is projected towards the two different CRL catalytic machineries.

We compared our structure showing CDO1 ubiquitylation by LRRC58 with neddylated CUL2-RBX1-ARIH1 with the prior structure representing substrate ubiquitylation by a neddylated CUL1-RBX1-ARIH1 complex[44] (Supplementary Fig. 13a). Remarkably, the catalytic ARIH1-ubiquitin portion of the prior structure with CUL1 fit well in the map representing this assembly ubiquitylating CDO1. The same catalytic architecture was also observed in published maps for another CUL1 complex[44], and another map for a complex representing

**Table 1 | Data collection, refinement and validation statistics**

| | Structure representing CDO1 ubiquitylation by LRRC58 with neddylated CUL5-RBX2-ARIH2 PDB: 9T7V EMD-55658 EMD-55659 EMD-55660 | Screening dataset map representing CDO1 ubiquitylation by LRRC58 with neddylated CUL2-RBX1-ARIH1 | Maps representing CDO1 ubiquitylation by LRRC58 with neddylated CUL2-RBX1-ARIH1 EMD-55652 EMD-55653 EMD-55654 EMD-55655 EMD-55656 | Low resolution dataset of CDO1-LRRC58 CRL2 complex | Low resolution dataset of CDO1-LRRC58 CRL5 ARIH2 complex |
|---|---|---|---|---|---|
| **Data collection and processing** | | | | | |
| Magnification | 105,000 | 22,000 | 105,000 | 105,000 | 22,000 |
| Voltage (kV) | 300 | 200 | 300 | 300 | 200 |
| Electron exposure (e–/Å²) | 50.2 | 60.0 | 51.0 | 54.0 | 60.0 |
| Defocus range (μm) | – 0.6 to – 2.2 | – 1 to – 2.6 | – 0.6 to – 2.2 | – 0.6 to – 2.2 | – 1 to – 2.6 |
| Pixel size (Å) | 0.8512 | 1.841 | 0.8512 | 0.8512 | 1.841 |
| Symmetry imposed | C1 | C1 | C1 | C1 | C1 |
| Initial particle images (no.) | 17,986,650 | 11,643,225 | 13,861,287 | 19,781,768 | 3,879,566 |
| Final particle images (no.) | 847,022 | 206,311 | 1,462,967 | 169,792 | 630,414 |
| Map resolution (Å) | 2.95 | 4.88 | 3.70 | 4.51 | 5.39 |
| FSC threshold | 0.143 | 0.143 | 0.143 | 0.143 | 0.143 |
| Map resolution range (Å) | 2.31 – 8.37 | 4.40 – 11.96 | 2.90 – 8.43 | 3.82 – 53.38 | 4.79 – 13.26 |
| **Refinement** | | | | | |
| Initial model used (PDB code) | AlphaFold3, 4N9F, 9SDX, 7B5M, | | | | |
| Model resolution (Å) | 2.95 | | | | |
| FSC threshold | 0.143 | | | | |
| Model resolution range (Å) | 2.31 – 8.37 | | | | |
| Map sharpening $B$ factor (Å²) | -60 | | | | |
| Model-map correlation (box) | 0.77 | | | | |
| Model composition | | | | | |
| Non-hydrogen atoms | 15047 | | | | |
| Protein residues | 1893 | | | | |
| Ligands | SY8: 1 ZN: 8 FE: 1 | | | | |
| $B$ factors (Å²) | | | | | |
| Protein | 90.97 | | | | |
| Ligand | 109.77 | | | | |
| R.m.s. deviations | | | | | |
| Bond lengths (Å) | 0.003 | | | | |
| Bond angles (°) | 0.552 | | | | |
| Validation | | | | | |
| MolProbity score | 1.95 | | | | |
| Clashscore | 5.82 | | | | |
| Poor rotamers (%) | 2.42 | | | | |
| Ramachandran plot | | | | | |
| Favored (%) | 95.06 | | | | |
| Allowed (%) | 4.94 | | | | |
| Disallowed (%) | 0.00 | | | | |

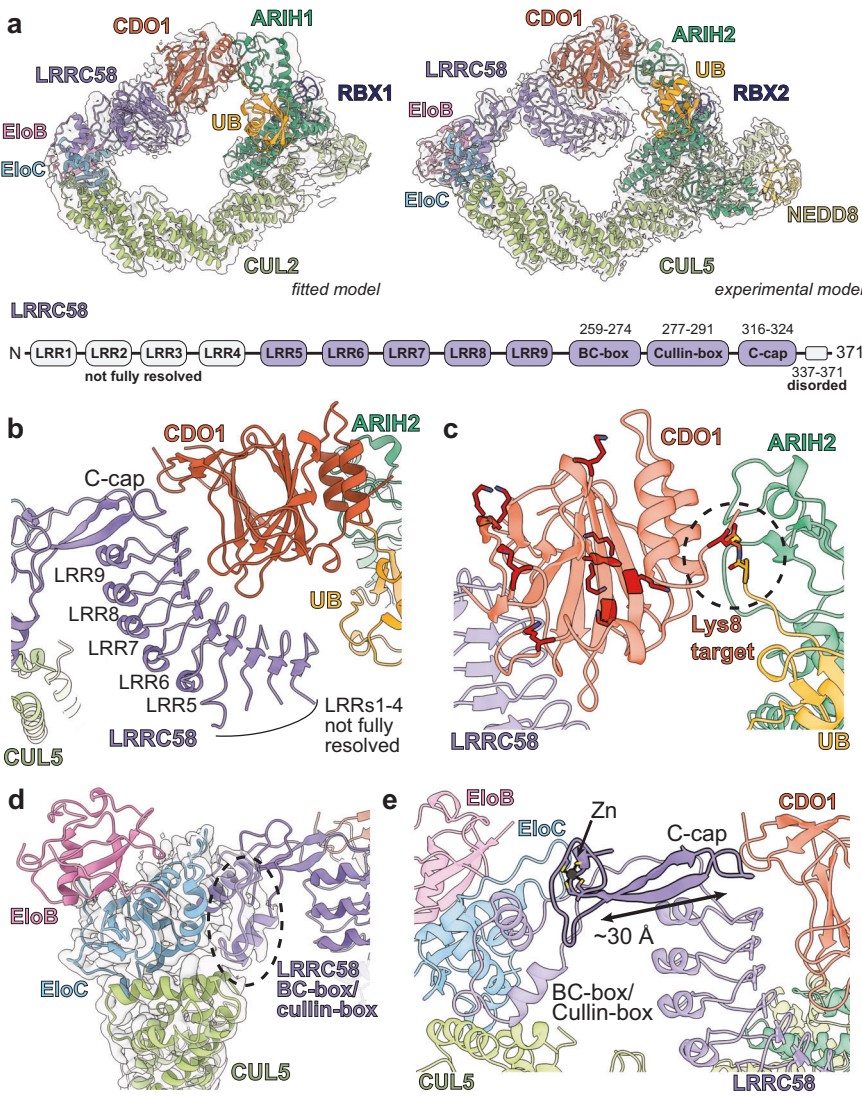

**Fig. 5 | Cryo-EM structure of CDO1 engagement by LRRC58-CRL explains selective substrate Lys targeting. a** The left panel displays a model of the sample representing CDO1 ubiquitylation by LRRC58 with neddylated CUL2-RBX1-ARIH1. Structural coordinates from EloB/C-CUL2 N-terminal domain (pink, light blue, and light green, respectively; PDB:8WQH), CUL2 C-terminal domain (light green; PDB:8Q7R), ARIH1 Ariadne domain (green) and RBX1 (navy; PDB:7B5M), CDO1-LRRC58 complex (orange and purple, respectively from CDO1-LRRC58 in complex with CUL5) and from AlphaFold3 models of and ARIH1-Ub (gold) were rigid-body fit into the corresponding composite EM density (transparent gray). The right panel displays the structure representing CDO1 ubiquitylation by LRRC58 with neddylated CUL5-RBX2-ARIH2 (EloB in pink, EloC in light blue, LRRC58 in purple, CDO1 in orange, NEDD8 in yellow, CUL5 in light green, Ub in gold, RBX2 in navy, ARIH2 in green) and the corresponding composite EM density (transparent gray). Electron density for NEDD8 is not apparent in the CUL2 complex, consistent with previous structures[43,44] due to the differing configurations of NEDD8 in CUL5-RBX2 versus CUL2-RBX1. The LRRC58 domain schematic is shown below structures. **b** Close-up of the interface between LRRC58 (purple) and CDO1 (orange) in the neddylated CRL5 complex. **c** Close-up of CDO1 (orange) engagement of the catalytic Rcat domain of ARIH2 (LRRC58 in purple, ARIH2 in green, Ub in gold). CDO1 Lys side chains are shown as opaque sticks. The Lys8 target position, here replaced by a cysteine-substitution to generate a stable mimic of the ubiquitylation intermediate to facilitate cryo-EM structure determination, is highlighted (black dashed circle). **d** Close-up of the EloC-LRRC58-CUL5 interface with cryo-EM density (transparent gray) for the EloC and N-terminal CUL5 subunits (EloC in light blue, EloB in pink, CUL5 in light green, and LRRC58 in purple). **e** Close-up of LRRC58's 2-stranded β-sheet from which emanate the cullin-binding regions (BC- and cullin-box), the Zinc binding loop (side chains of Zinc-coordinating cysteine residues shown), and the C-cap region (LRRC58 in purple, CUL5 in light green, EloC in light blue, EloB in pink, CDO1 in orange).

ubiquitylation by KHLDC10-CUL2[74] (Supplementary Fig. 13b, c). These data demonstrate that various neddylated RBX1-based CRLs can mediate substrate ubiquitylation by a generalizable ARIH1-ubiquitin catalytic architecture.

Neddylated CUL5-RBX2-ARIH2 complexes also show a generalizable core assembly. Although there is no prior structure representing substrate ubiquitylation by neddylated CUL5-RBX2-ARIH2, there is precedent for such an assembly that did not have ubiquitin at the ARIH2 active site[62]. In the prior structure, the ARIH2 catalytic domain was not visible. Nonetheless, the neddylated CUL5-RBX2-ARIH2

components that were visible superimpose with the LRRC58-CDO1 complex (Supplementary Fig. 13d). Thus, these data suggest a subtly different catalytic arrangement for ubiquitin transfer by CUL5-ARIH2 compared to CUL2-ARIH1 (Supplementary Fig. 13e, f), although the functional consequences of these distinctions will depend on future studies.

## Discussion

Our data, together with work of others[52–54] provide a molecular pathway for elegant regulation, discovered decades ago, through

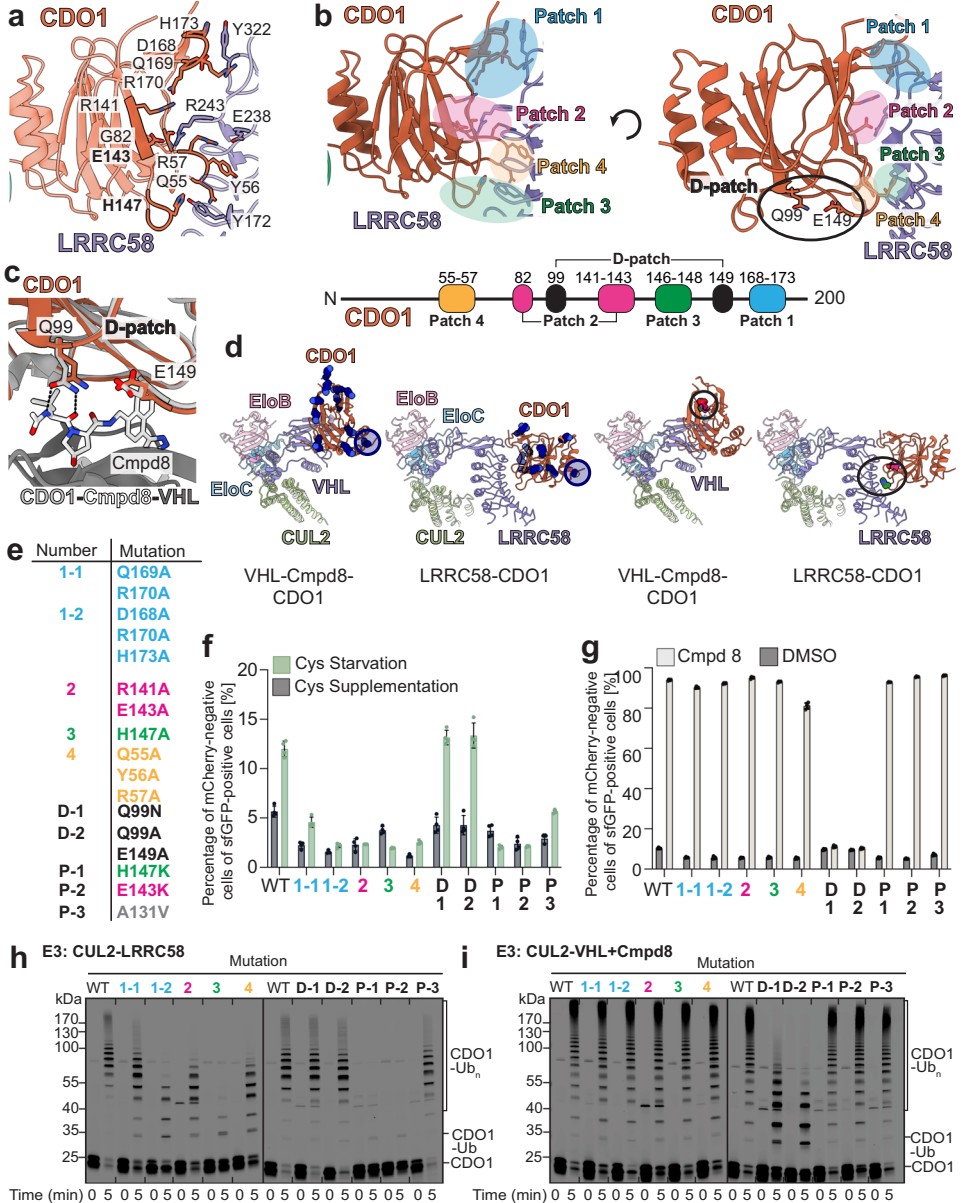

**Fig. 6 | Perturbation of the LRRC58-CDO1 interface decreases Cys-regulated but retains orthogonal targeted protein degradation of CDO1. a** Residues along the LRRC58-CDO1 interface of the structure representing CDO1 (orange) ubiquitylation by LRRC58 (purple) with neddylated CUL5-RBX2-ARIH2 are labeled and shown in full opacity. Residues found mutated in patients are labeled in bold. **b** The LRRC58-CDO1 interface is subdivided into patches. D-patch residues, involved in degrader-induced interaction with VHL, are shown with respect to the interface patches on the right. CDO1 patch schematic shown below. **c** CDO1-Cmpd8-VHL-EloB/C crystal structure (PDB:8VL9, CDO1 light gray, VHL dark gray, Compound 8 (Cmpd8) white) aligned to CDO1 (orange) of the sample representing CDO1 ubiquitylation by LRRC58 with neddylated CUL5-RBX2-ARIH2. Hydrogen bonding interactions are shown (black dotted lines). **d** Cmpd8-mediated recruitment of CDO1 to VHL orients CDO1 differently than LRRC58, shown by alignment of a chimeric structure of VHL-Cmpd8-CDO1 (PDB:8VL9) in complex with CUL2^NTD-EloB/C (PDB:4WQO) to the fitted model of the LRRC58-CDO1-CUL2 complex. On left, lysines are colored as dark-blue spheres; Lys8 position is highlighted in a dark-blue circle. Sites of patient

mutations, H147 and E143 (green and pink spheres, respectively), are highlighted with black circles, on right. **e** Summary of CDO1 mutations used. **f** Reporter stability for CDO1 variants expressed in HEK293T cells grown in complete or cysteine-free media. All mutations made to CDO1 patches lose sensitivity to cysteine starvation, whereas D-patch mutations respond. **g** Reporter stability as in (**d**) except comparing cells gown in complete media and treated with Cmpd8 or DMSO. All CDO1 variants are efficiently degraded, including the patient variants, except for D-patch mutations. **h** In vitro reconstituted assays comparing WT and mutant Cy5-CDO1 ubiquitylation by neddylated LRRC58-CUL2. Mutations only in patch interfaces, including patient variants, reduced CDO1 ubiquitylation. **i** Same as in (**h**) except with neddylated VHL-CUL2 and Cmpd8. CDO1 patch mutants, and all patient variants, show WT-like ubiquitylation efficiencies, but D-patch mutants display reduced ubiquitylation. In (**f**) and (**g**), the average values from $n = 4$ independent replicates are shown with error bars reporting standard deviation. Fluorescence scans in (**h**) and (**i**) are representative of $n = 2$ technical replicates. Source data provided as Source Data file.

degradation of CDO1 by the ubiquitin-proteasome system when cysteine is limiting[35–37]. Our proteomics-based strategy complements previous genetic studies in worms, system metabolomics across mammalian systems, and CRISPR screening in human cells expressing a stability reporter[52–54] that also identifies co-regulation of a cullin-RING

ligase receptor, LRRC58, and CDO1 as its substrate (Figs. 1, 2). Cysteine starvation protects LRRC58 from ubiquitin-mediated degradation (Figs. 1, 2). This in turn controls CDO1 through its direct ubiquitylation and degradation (Fig. 2).

Through cellular and biochemical reconstitution, our study uniquely defines facets underlying CDO1 regulation by LRRC58 and by the recently developed molecular glue degrader[55] Cmpd8 with VHL. While the endogenous LRRC58-dependent pathway preferentially targets CDO1 Lys8 for ubiquitylation, the degrader molecule overrides this through ubiquitylation on multiple sites (Fig. 4). Cryo-EM data support that LRRC58 binds a different side of CDO1 in comparison with the VHL/Cmpd8 complex (Fig. 6b–d). Importantly, our structural data demonstrate that LRRC58 projects CDO1 Lys8 towards a CRL ubiquitylation active site. Although future structural studies are required to show how the VHL/Cmpd8-bound CDO1 undergoes ubiquitylation, it is notable that this complex could allow access to other surface lysines (Fig. 6d). These functional differences highlight how distinct presentations of a single substrate to a common cullin can tune ubiquitylation and degradation efficiencies.

Distinctions in endogenous versus drug-mediated substrate ubiquitylation is of relevance towards successful development of therapeutics mediating targeted protein degradation. For instance, we note that two of three CDO1 mutations recently reported as the only genetic etiology discovered in patients with a rare neurological disorder (E143K and H147K) map to CDO1 patches 2 and 3 contacting LRRC58 (Fig. 6a, d, e)[73]. The data here show that these mutations are defective for LRRC58-dependent ubiquitylation. Ubiquitylation was not affected by the other patient mutation, A131V, which is not found on a LRRC58-interacting CDO1 patch but instead is proximal to the CDO1 active site and may impact catalytic activity (Fig. 6h). Importantly, our data show that all three patient variants are targeted by VHL/Cmpd8 (Fig. 6i). Thus, our data indicate that disease-associated proteins could be targeted by orthogonal recognition induced by degrader molecules.

Our structural data also revealed unexpected plasticity for LRRC58's cullin-box, which can bind CUL5 as well as CUL2. This concept is consistent with our finding that knockdown of both CUL2 and CUL5 is required to counteract the cysteine starvation-dependent decrease in endogenous CDO1. CUL2 may play a dominant role in at least some settings, as its knockdown was sufficient to stabilize our exogenously-expressed CDO1 reporter. These findings are in line with others that show potential partnering of LRRC58 with CUL2 or CUL5 in cells[52–54,65]. Future studies in animal models will be required to dissect cullin-specificity of physiological regulation. Nonetheless, in combination with previous work, the structures elucidated here define a consensus active configuration for ARIH1-mediated ubiquitylation of substrates recruited to neddylated CUL1 and CUL2 E3s, and a twist in ARIH2 for the corresponding ubiquitylation complex with substrate recruited to a neddylated CUL5 E3 (Supplementary Figs. 13e, f, 15).

While our manuscript was in preparation, two other studies also reported human LRRC58 regulation of CDO1. The confluence of independent data supports the robustness of our findings and highlights the range of methodologies available to discover metabolic signaling through the CRL pathways. One study relied on an elaborate approach - correlating levels of 285 metabolites with 11,868 proteins across genetically diverse mouse and human samples - to discover a linkage between taurine and hypotaurine with LRRC58 and CDO1[54]. The other study employed ubiquitin-focused CRISPR screening. This preprint reported the role of LRRC58 in destabilizing a fluorescent CDO1 stability reporter under cysteine-limiting conditions (and in destabilizing a stability reporter for LRRC58 itself in cysteine-rich conditions)[52]. The orthologous *C. elegans* pathway was also recently reported[53], showing that LRRC58 controls sulfur metabolism through post-translational regulation of CDO1.

By contrast, while our initial results relied on proteomic strategies we developed to reveal responses to signals[38], our follow-up experiments showed that the relationship between levels of cysteine, LRRC58 and CDO1 were apparent by straightforward proteome analysis by DIA-MS combined with high-throughput computational modeling.

Retrospectively, our pipeline is consistent with prior findings that many CRL substrate receptors are regulated by autodegradation that is suspended when these E3s are needed to ubiquitylate substrates[38,46,47,56–58]. Our study further highlights how proteomics can shed light on E3 targets, through comparison of total proteomes and modeling of protein complexes. This strategy can be universally applied across cell lines and perturbations, and it yields data on endogenous proteins without requiring sophisticated metabolomics, elaborate computation, or CRISPR screening to elucidate pathways.

Now, CDO1 joins IDO1 (indoleamine 2,3-dioxygenase 1) and TDO2 (tryptophan 2,3-dioxygenase) as oxidoreductase enzymes that are downregulated by a CRL in response to metabolic stress[75–77]. IDO1 is destabilized when heme biosynthesis is inhibited. Recent studies showed how the level of IDO1 is controlled in a heme-dependent manner by the KLHDC3-CUL2 E3[76]. KLHDC3-CUL2 preferentially ubiquitylates a lysine near the C-degron of apo-IDO1. However, this IDO1 region is reshaped upon heme binding, allosterically hindering accessibility to the CRL. Meanwhile, the abundance of TDO2, like CDO1, is regulated by the availability of its amino acid substrate[78]. Degradation of these enzymes when their respective tryptophan or cysteine substrate is limiting maintains homeostasis by preventing toxic depletion of a crucial amino acid. TDO2 is stabilized by it directly binding tryptophan at an "exosite" distal from the active site[77,79]. When tryptophan is limiting, TDO2 is subject to proteasomal degradation mediated by a two-step CK2 kinase and FBXW11-CUL1 cascade[77].

A distinct mode of metabolite sensing is now suggested from the inhibition of LRRC58-mediated degradation of CDO1 when cysteine is plentiful. However, the cysteine-sensing mechanism remains unclear. LRRC58 residues serving as a "cysteine sensor" were identified by saturation mutagenesis of an LRRC58 stability reporter described in a recent preprint[52]. However, mapping those residues onto our structure of the LRRC58-CDO1 complex did not reveal an amino acid-binding motif (Supplementary Fig. 16). On the other hand, the structure shows that some "cysteine sensor" residues promote LRRC58 binding to CDO1, raising the possibility that substrate binding protects LRRC58 from degradation. Such regulation does not explain how cysteine levels are perceived at a molecular level. Other "cysteine sensor" residues map to a cysteine-rich Zn-binding element, which orients LRRC58's substrate relative to the cullin and ubiquitylation active site (Supplementary Fig. 16). These residues could potentially play a role in projecting LRRC58 for ubiquitylation, although deciphering the cysteine sensing mechanism will require further investigation. Irrespective of the molecular mechanism, the regulation of CDO1 through metabolic control of LRRC58 stability (Figs. 1, 2) provides yet another means for the regulation of an oxidoreductase enzyme by its amino acid substrate. It seems likely that widespread control of metabolic enzymes through the ubiquitin-proteasome system awaits discovery. We anticipate that crosstalk between the ubiquitin system and metabolism will be unearthed by a plethora of technologies - genetics[53], multi-sample multiomics[54], CRISPR screening[52], and proteomics together with structural modeling (Figs. 1, 2) - just as these strategies all converged to decipher the LRRC58 response to cysteine abundance.

## Methods

### Materials
All chemicals are from Sigma-Aldrich, unless otherwise stated.

### Cloning, protein expression and purification
All plasmids in this study were generated using Gibson Assembly. Plasmid sequences were verified through Sanger sequencing (Microsynth Seqlab). LRRC58 was cloned into a modified pFastBac vector carrying an N-terminal TwinStrep-tag with a 3C cleavage site. EloB/C were cloned untagged into a pFastBac vector. Coexpression was performed in *Trichoplusia Ni* High Five insect cells. Cell pellets were resuspended in 50 mM Tris, pH 8.0, 200 mM NaCl, 5 mM DTT,

supplemented with 100 U/mL smDNAse, 1 μM pepstatin, 10 μg/mL aprotinin, 5 μg/mL leupeptin, 1 mM AEBSF, 5 mM sodium fluoride, 5 mM sodium orthovanadate, 5 mM β-Glycerophosphate, 5 mM sodium pyrophosphate, 1X PhosSTOP Phosphatase Inhibitor Cocktail (Roche), and 2.5 mM MgCl before lysis by sonication on ice. Lysates were cleared via centrifugation before incubation with Strep-Tactin Sepharose resin (IBA Lifesciences) for 30 min. Resin was washed with 50 mM Tris pH 8.0, 150 mM NaCl, 2 mM DTT and the protein was cleaved off of the beads via an overnight incubation with ~1:50 w/w HRV3C protease at 4 °C. The protein was further purified via anion exchange, eluting along a gradient between 50-1000 mM NaCl in 50 mM Tris pH 8.0, 2 mM DTT, and subsequent size exclusion chromatography (SEC) using a Superdex 200 Increase 10/300 GL column into SEC Buffer 25 mM HEPES pH 7.5, 150 mM NaCl, 0.5 mM TCEP.

WT CDO1 and mutants were cloned into a pGEX4T1 vector and expressed as GST-fusion proteins with a TEV cleavage site in *E. coli* Rosetta cells. Expression was induced using 200 μM Isopropyl β-D-1-thiogalactopyranoside (IPTG) along with 200 μM $FeSO_4$. Purification was performed using GST-affinity chromatography with subsequent TEV cleavage, followed by anion exchange and final SEC into SEC buffer. The active CRL-binding Fab, N8C_Fab3b, was cloned, expressed, purified, and biotinylated as previously described[38]. CUL5-NTD(2-386) and E2s UBE2R2 and UBE2L3 were expressed as GST-fusions in *E. coli* Rosetta. VHL-EloB/C was coexpressed using a pGEX4T1 GST-TEV-VHL (residue 54 to C-term) construct and a pACYCDuet-1 vector carrying untagged Elongin B and Elongin C in *E. coli* BL21(DE3) cells. UBA1 and CUL2-NTD (1-380,V340D,L344K,V376D) was expressed as GST-TEV fusion in insect cells. GST-fusion proteins were purified by GST-affinity chromatography followed by TEV cleavage, anion exchange and SEC into SEC buffer. ARIH1, both the WT and open variant (F430A, E431A, E503A) were expressed and purified as described previously[43]. ARIH2, both the WT and open variant (L381A, E382A, E455A), were expressed and purified as described previously[62]. Neddylated CUL5 and CUL2 and components for neddylation reaction were expressed and purified as well as subsequent neddylation reactions were performed as described previously[46,62]. WT and $K_0$ ubiquitin was expressed, purified and, where applicable fluorescently labeled with Fluorescein as described previously[69]. SortaseA was cloned into a modified pRSF vector carrying N-terminal MBP-TEV and C-terminal His-Tag and expressed in *E. coli* Rosetta cells. Purification proceeded through His-affinity chromatography with subsequent IEX and SEC into the final SEC buffer. All proteins were verified using intact mass at the MPIB core facility.

### Fluorescent labeling of CDO1
WT CDO1 and mutants were N-terminally labeled using Cy5-GSGGLPETGG peptide obtained from the MPIB Biochemistry core facility. Reactions were performed at room temperature using 20 μM CDO1 variant, 3 μM SortaseA, 400 μM Cy5-peptide in reaction buffer (50 mM TRIS pH 8,150 mM NaCl, 10 mM $CaCl_2$). Reaction were passed over amylose resin (NEB) to remove SortaseA, and then labeled CDO1 was further purified into SEC buffer using a Superdex 75 Increase 10/300 GL column.

### Cell culture
All cells were cultured with 10% fetal bovine serum (Gibco), 100 units/mL penicillin and 100 μg/mL streptomycin at 37 °C and 5% carbon dioxide. Dulbecco's Modified Eagle Medium (DMEM, Gibco) was used, unless otherwise stated. HEK293T (ACC 635) and HeLa (ACC 57) cells were obtained from the Deutsche Sammlung von Mikroorganismen und Zellkulturen (DSMZ). Jurkat (TIB-152), HepG2 (HB-8065), and SKBR3 (HTB-30) cells were purchased from the American Type Culture Collection (ATCC). All cell lines were routinely tested for Mycoplasma contamination using the MycoStrip Mycoplasma Detection Kit (InvivoGen) or a PCR Mycoplasma detection kit (Applied Biological

Materials). HEK293T, HeLa, and SKBR3 cells were cultured in DMEM supplemented with 10% fetal bovine serum (FBS) and penicillin–streptomycin (P/S). HepG2 (adapted to grow in RPMI media) and Jurkat cells were maintained in RPMI 1640, GlutaMAX (Gibco) supplemented with 10% FBS, 1 × MEM NEAA (Gibco) and P/S. SKBR3 cells were additionally cultured in McCoy's 5 A medium (Gibco, 16600082) supplemented with 10% FBS and P/S. All experiments were performed using cells passaged fewer than 30 times from the original source. For cysteine manipulation, HEK293T, HeLa, and SKBR3 cells were seeded in 6-well plates at 1×10⁶ cells per well (in quadruplicate) in cysteine- and methionine-deficient DMEM (Gibco, 21013024) supplemented with 1 × GlutaMAX, 1 × MEM NEAA, 1 × sodium pyruvate (Gibco), 0.201 mM L-methionine, and 0.402 mM L-cysteine. HepG2 and Jurkat cells were cultured under analogous conditions using cysteine- and methionine-deficient RPMI (Sigma, R7513) with the same supplements. The following day, cells were washed with PBS (Gibco) and media replaced with either starvation medium, complemented medium, or 10 × cysteine medium. Starvation medium lacked L-cysteine entirely, whereas 10 × cysteine medium was supplemented with 4.02 mM L-cysteine. After 24 h, cells were washed twice with PBS and snap-frozen in liquid nitrogen for further analysis.

### Amino acid starvation release assay (L-cystine)
Cells were plated in triplicate (IP or for total proteome) in 6-well plates at 1 × 10⁶ cells/well one day prior to the start of the assay. The next day, all wells were washed once in room temperature PBS and the media replaced according to assay groups with complete DMEM (unstarved control) or starvation media DMEM lacking L-cystine (Thermo Fischer Scientific 21013-024 media lacking amino acids L-glutamine, L-methionine and L-cystine and supplemented with dialyzed FCS, 4 mM L-glutamine and 0.201 mM L-methionine, P/S (from 100X stock) and 2 g/L glucose final concentration). After 22–24 h, three groups of starvation media cultures were washed once with normal DMEM (containing all amino acids and undialyzed FCS with 10 μM MG132 (MedChemExpress) and 10 μM CB5083 (MedChemExpress)) while the starved and unstarved controls were lysed for IP or total proteomes, as below. The "released" cultures that were starved and now incubated in complete media were lysed at 30, 60 or 120 min in complete media. Therefore, there are five experimental groups: unstarved control, starved control and "released" cultures at 30, 60 and 120 min in complete media. Lysis of the IP samples was accomplished by first gently washing and then incubating with room temperature PBS containing 2 μM MLN4924 (MedChemExpress) and 2 μM CSN5i (MedChemExpress) for three minutes without agitation. Cells were then washed gently with PBS lacking inhibitors. Cells were then lysed on ice with 500 μL 25 mM HEPES pH 7.5, 5% glycerol, 150 mM NaCl, 0.5% NP-40, 1x HALT protease/phosphatase inhibitor (Thermo Fisher Scientific), 2 μM MLN4924 (MedChemExpress), and 2 μM CSN5i-3 (MedChemExpress). IP samples were scraped and centrifuged for 7 min at 14,000 × *g* and the clarified supernatant transferred to a new tube on ice and flash frozen in liquid nitrogen. For total proteomes, the same procedure was used but with 500 μL 60 mM TEAB pH 8.5, 10 mM TCEP, and 25 mM 2-chloroacetamide.

### IP-MS
Active CRL IP-MS was performed as described in Henneberg, et al.[38], briefly described below, with the following changes. Frozen lysates were thawed or HEK293T frozen cell pellets were resuspended and lysed in 25 mM HEPES pH 7.5, 5% glycerol, 150 mM NaCl, 0.5% NP-40, 1x HALT protease/phosphatase inhibitor (Thermo Fisher Scientific), 2 μM MLN4924 (MedChemExpress), and 2 μM CSN5i-3 (MedChemExpress), cleared by centrifugation at 18,000 × *g* for 3 min at 4 °C and filtered through a 0.22 μM cellulose acetate spin filters (Corning). Four biological replicates were prepared for mass spectrometry. High-Capacity Magne Streptavidin Beads (Promega) were coated with N8C_Fab3b,

following the manufacture's protocol. The equivalent of 10 μL of Fab-coated bead slurry were added to each lysate and incubated while rotating for 30 min at 4 °C. Beads were washed twice with wash buffer A (25 mM HEPES pH 7.5, 5% glycerol, 150 mM NaCl, 0.5% NP-40), twice with wash buffer B (25 mM HEPES pH 7.5, 5% glycerol, 150 mM NaCl), and twice with wash buffer C (25 mM HEPES pH 7.5, 150 mM NaCl). After the final wash, beads were resuspended in 0.1% trifluoroacetic acid (TFA) and incubated at room temperature for 5 min with shaking. The beads were removed before neutralizing in 66 mM triethylammonium bicarbonate buffer (TEAB) (Merck). 2-Chloroacetamide was added to a final concentration of 40 mM and TCEP (Serva) was added to a final concentration of 10 mM before incubating for 5 min at 45 °C, with shaking. Overnight digestion at 37 °C, with shaking, was done with 0.5 μg of both Trypsin and LysC (FUJIFILM Wako) per sample. The following day, the peptide concentration was determined through a BCA assay (Thermo Fisher Scientific) or a tryptophan fluorescence assay[80].

## Total proteome analysis
For total proteome analysis, the frozen resuspensions were thawed or frozen cell pellets were resuspended 60 mM TEAB pH 8.5, 10 mM TCEP, and 25 mM 2-chloroacetamide. Four biological replicates were prepared for mass spectrometry. Lysates were incubated at 98 °C for 5 min before sonicating using a Bioruptor (Diagenode) at 15 cycles of 30 s on and 30 s off at the low setting. Samples were incubated again at 98 °C for 5 min before centrifuging at 18,000 × g for 20 min at 4 °C. Cleared lysates were digested overnight at 37 °C with both Trypsin and LysC at 1:100 w/w, with shaking. Digested peptide concentration was determined through a BCA assay.

## LC-MS/MS measurements
200 ng of each sample was loaded onto Evotips Pure (Evosep) according to the manufacturer's instructions. Samples were run on an Evosep One LC unit (EvoSep, EV-1000) connected to a TimsTOF Pro 2 mass spectrometer (Bruker Daltonics) via a CaptiveSpray ion source with a 10 μm fused silica inner-diameter emitter (Bruker Daltonics, 1865691). A 30 sample per day program was used to perform chromatographic separation with a mobile phase system comprised buffer A (0.1% formic acid in water) and buffer B (0.1% formic acid in acetonitrile) on a 15 cm × 150 μm column packed with 1.9 μm C18 beads (Bruker Daltonics, 1893471), which was held at 50 °C. A 20 scan dia-PASEF workflow was used for data acquisition, with each scan incorporating two ion-mobility windows, to cover the m/z range 350–1200. Isolation window widths were fixed using the *py_diaid* tool[81]. The ion accumulation and ramp phases were set at 100 ms, and the ion mobility was set between 0.7 and 1.3 V s cm$^{-2}$. A linear collision energy gradient was used, starting from 20 eV at $1/K_0 = 0.6$ V s cm$^{-2}$ and reaching 59 eV at $1/K_0 = 1.6$ V s cm$^{-2}$.

## MS data analysis
Precursor and fragment identification was performed using DIA-NN 1.9.2 in library-free mode[82], searching against the reviewed human proteome (Uniprot, November 2024, 20,663 entries without isoforms). The DIA-NN results are provided in Supplementary Data 1. Cleavage sites at lysines and arginines were allowed, with 1 missed cleavage and a minimum and maximum peptide length of 7 and 30, respectively. A maximum of two variable modifications were allowed, and variable modifications for methionine oxidation, N-terminal acetylation, and cysteine carbamidomethylation were enabled. MBR and "deep-learning-based spectra, RT and IM prediction" were also allowed. Using Python (3.13.5) and the packages pandas (2.3.0) and directlfq (0.3.2), protein intensities were normalized using directLFQ[83]. DirectLFQ normalized protein intensities in each replicate were plotted for specific identified protein groups, and the mean and standard

deviation were visualized using Prism (10.6.1). DirectLFQ normalized protein intensities are also reported in Supplementary Data 1.

Data were also analyzed using the Perseus software package[84] (2.0.9.0). Protein intensities were log2-transformed, and the datasets were filtered to contain no missing value in at least one experimental condition for all of the protein groups. Missing values were by default imputed using a normal distribution with a width of 0.3 and a down-shift of 1.8. Perseus (2.0.9.0) was used to perform t-tests (two-sided, permutation-based FDR calculation, 0.05 FDR, 250 randomizations, s0 = 0.1) and generate volcano plots with 5% FDR threshold curves which were visualized using Python (3.13.5) and the packages pandas (2.3.0), numpy (2.2.6), and plotly (6.1.2), with code generation assistance from Claude 4.5 Sonnet (Anthropic). The values plotted in volcano plots and bar charts are provided as a Source Data file. CRL substrate binding modules and the cullins they are assumed to primarily associate with, referred to during data processing and visualization, are provided in Supplementary Data 1.

## HT-Colabfold
From the total proteome analyses comparing the complete media proteomes and excess cysteine proteomes; a list of all proteins that were not detected in the normal proteome but only were detected in the cysteine excess proteome was curated for each cell line. The sequences of each protein (provided in Supplementary Data 2) were used as input using a local High-Throughput Implementation of Colabfold (HT-Colabfold) developed by Hohmann, et al.[59]. The sequences of LRRC58, EloB, and EloC were used as bait. EloB and EloC were included because LRRC58 is insoluble without this co-expression. HT-ColabFold ran with a 95% positive match rate. The output IPTMavg scores were plotted against the inverse of the PEAKavg scores using Prism (10.6.1).

## Sample preparation for RNA-seq
As described above the amino acid starvation treatment was performed on HEK293T cells. Cells were lysed using 1 mL of TRIzol™ (Thermo Fisher Scientific) and the samples were stored at −80 °C overnight. Subsequently, 200 μL of chloroform was added, and the RNA phase was collected following centrifugation for 10 min at 4 °C at 13000 x g. The RNA fraction was precipitated with 500 μL of 2-propanol and washed with 1 ml of EtOH 70%. Finally, the pellet was left to dry and resuspend in 25 μL of RNAse-free water.

## Bulk mRNA sequencing
Bulk mRNA sequencing libraries were prepared with 1 μg of total RNA of each sample using the NEBNext Ultra™ II Directional RNA Library Prep Kit for Illumina® (E7760, NEB) with NEBNext® Poly(A) mRNA Magnetic Isolation Module (E7490, NEB), according to the standard manufacturer's protocol. Total RNA and the final library quality controls were performed using Qubit™ Flex Fluorometer (Q33327, Thermo Fisher Scientific) and 4200 TapeStation System (G2991BA, Agilent) before and after library preparation. The libraries were sequenced on Element AVITI (2 × 75 bp, on average 20 M reads per sample) and demultiplexed by bases2fastq software (Element).

## Next-generation sequencing data analysis
**Read processing and quantification.** Initial quality control of the raw sequencing data was performed with FastQC (v.0.11.7)[85]. Subsequently, high-quality reads were mapped to the human reference genome (build GRCh38, Ensembl) using the STAR aligner (v. 2.7.10b)[86] with its default parameters. To generate a gene-level count matrix, the aligned reads were assigned to genomic features defined by Ensembl annotations using the featureCounts tool (v. 2.0.4)[87,88]. During this step, reads overlapping multiple genes were discarded to ensure unambiguous quantification.

**Differential gene expression analysis.** We identified differentially expressed genes (DEGs) using the DESeq2 package[89,90] in the R statistical environment (v. 4.3.2)[91]. The raw count data was normalized for library size using DESeq2's internal median-of-ratios method. Prior to statistical testing, the dataset was filtered to remove low-abundance genes, keeping only those with a count of 10 or more in a minimum of three samples. Differential gene expression between the two experimental conditions was assessed by estimating gene-wise dispersion and fitting a negative binomial generalized linear model. A stringent threshold of an adjusted p-value (Benjamini-Hochberg procedure) ≤ 0.01 was applied to define statistically significant differentially expressed genes for further investigation. Values were plotted using Python (3.13.5) and the packages pandas (2.3.0), numpy (2.2.6), matplotlib (3.10.3), and seaborn (0.13.2) with code generation assistance from Claude 4.5 Sonnet (Anthropic).

### Lentivirus packaging and transduction
Reporter cell lines were established via lentiviral transduction. For lentivirus packaging, $0.5 \times 10^6$ HEK293T cells were seeded per well of a 6-well plate. The next day, the plasmids psPAX2 (1 μg), pVSV-G (0.4 μg) and lentivirus transfer plasmid (1.5 μg) were transfected using Xtreme Gene9 (Roche) according to the manufacturer's instructions. Lentivirus was harvested after 48 h. For lentivirus transduction, $1 \times 10^6$ target cells per well of 6-well plate and the addition of undiluted lentivirus and polybrene transfection reagent (Merck) at a final concentration of 8 μg/ml were used. After 2 days, the media with lentivirus was removed, and the cells were washed with 1 x DPBS. Fresh medium was added to the cells.

### Cell-based stability reporter assay
Stability reporter expressing cells were seeded in Poly-D-Lysine (Gibco) coated 24-well plates in quadruplicates at $0.2 \times 10^6$ cells per well in complemented cysteine- and methionine-deficient DMEM supplemented with 2.01 mM L-cysteine. The next day, cells were washed with PBS before changing the media to starvation or complemented media. Additional treatments included 10 μM MG132 (MedChemExpress), 1 μM MLN4929 (MedChemExpress), 50 nM Compound 8 (NVS-VHL720 -MedChemExpress) or DMSO (D2438) as detailed in the figure legends for specific experimental conditions. After 24 h, cells were washed with 1 x DPBS and prepared for flow cytometry.

### Flow cytometry
Analytical flow cytometry was performed on an Attune NxT flow cytometer (Thermo Fisher Scientific) equipped with an automated plate reader. Cells were prepared by single-cell dissociation using TrypLE, then resuspended in 1× DPBS supplemented with 2 mM EDTA and DAPI staining solution (Miltenyi Biotec) for live/dead discrimination. GFP was detected using a 488 nm laser using the 530/30 filter, DAPI using a 405 nm laser with the 440/50 filter, and mCherry using a 531 nm laser with the 620/15 filter. Single cells were identified via FSC-A/SSC-A gating followed by doublet exclusion using FSC-A/FSC-H. Live cells were subsequently gated based on DAPI negativity. GFP-positive cells were then analyzed for mCherry. The complete gating strategy is shown in Supplementary Fig. 3. Flow cytometry data were processed using FlowJo v10.8.2, and statistical analyses were performed in GraphPad Prism v10.4.2.

### siRNA Knockdown
For siRNA-mediated knockdown, ON-TARGETplus siRNAs targeting CUL5 (Horizon, L-019553-00-0005), CUL2 (Horizon, L-007277-00-0005), and LRRC58 (Horizon, L-023580-01-0005), as well as a non-targeting control siRNA (Horizon, D-001810-10-20), were resuspended in 1× Dharmacon siRNA Buffer (Horizon). Reverse transfection was performed using Lipofectamine RNAiMAX Transfection Reagent (Thermo Fisher Scientific) according to the manufacturer's instructions, consisting of an initial 48 h reverse transfection followed by a second 24 h reverse transfection with the corresponding siRNAs.

### CRISPR Knockout
The LRRC58 CRISPR knockout (KO) cells were generated using a protocol described previously[92].

Two guides, targeting LRRC58 exon 2 and 3, respectively, were selected with the CRISPick Tool[93,94] to design primers to generate the following guide containing vectors: guide 1: AAAGTTGAGGAGG-TATGCTT (Forward primer: CACCGAAAGTTGAGGAGGTATGCTT, Reverse primer: AAACAAGCATACCTCCTCAACTTTC), guide 2: AGGA AGATCATAGGGAGTGT(Forward primer: CACCGAGGAAGATCATA GGGAGTGT, Reverse primer: AAACACACTCCCTATGATCTTCCTC). SgRNAs were dephosphorylated and annealed using 1 μL of each sgRNA (100 μM), 1 μL T4 ligase buffer (10x), 1 μL T4 Polynucleotide Kinase (Thermo Fisher Scientific), and 10 μL dH2O in a thermocycler with the following program: 37 °C for 30 min, 95 °C for 5 min, ramp down to 25 °C at 5 °C/min. Phosphorylated and annealed oligos were diluted 1:200 in ddH2O. Each sgRNA was cloned into the pSpCas9 vector using 100 ng pSpCas9 vector, 2 μL diluted oligos, 2 μL 10x TANGO buffer (Thermo Fisher Scientific), 1 μL 10 mM DTT, 1 μL 10 mM ATP, 1 μL Bbs1 (Thermo Fisher Scientific), 0.5 μL T4 Ligase (NEB), and 20 μL dH2O. The ligation reaction was incubated for six 2-step cycles at 37 °C for 5 min then 21 °C for 5 min. 11 μL of ligation reaction was incubated with 1 μL PlasmidSafe exonuclease, 1.5 μL 10 mM ATP and 1.5 μL PlasmidSafe buffer (10x) at 37 °C for 30 min, followed by 70 °C for 30 min. The treated plasmid was transformed into *E. coli* DH5α and selected on LB-Amp plates. Individual colonies were grown in 2 mL LB-Amp, and plasmids were isolated by MiniPrep (Qiagen) and sgRNA insertion was validated by sequencing. HEK293T cells were seeded in 24-well plates with a density of 0.1 million cells/well in 0.5 mL media/well. The following day the cells were transfected using Metafectene (Biontex) in two dilutions: 500 ng DNA and 2 μL Metafectene, each diluted in 30 μL OptiMEM (Gibco) and 1 μg DNA and 4 μL Metafectene, each diluted in 30 μL OptiMEM (Gibco). Diluted DNA was mixed with diluted Metafectene and incubated for 15 min before dropwise addition to cells. 72 h after the transfection, the cells were selected with 1 μg/mL Puromycin for three consecutive days. Individual clones of selected cells were seeded by FACS (Beckman Colter Cytoflex SRT) in 96-well plates and expanded.

Successful knockout clones were verified by genomic sequencing of the target locus. Briefly, the genomic DNA was extracted from 0.5-1 million cells of LRRC58 knockout clones of HEK293T using DNeasy Blood & Tissue Kit from Qiagen (Cat no. / ID. 69504) following the manufacturer's protocol. Further 50 ng of genomic DNA was used as template for PCR amplification of target loci using the following pair of primers, 5'-ACAGGATGAGACTTCCTGGGT-3' & 5'-CCTAATACCATTTA TGGTTTGTCTC-3'. The PCR product was cloned into the pCR-Blunt II-TOPO vector using Zero Blunt TOPO PCR Cloning Kit (Invitrogen, Cat no. K280002) and verified by Sanger sequencing using M13 standard primer (5'- TGTAAAACGACGGCCAG-3') and aligned to the LRRC58 target locus (shown in Supplementary Fig. 2).

### Transient transfection of LRRC58
For transient expressions, N-terminally Flag-tagged LRRC58 (both wildtype & A266F mutant) was cloned in pEG expression vector using standard molecular biology techniques. Constructs were verified by DNA sequencing. HEK293T cells were seeded in a 6-well plate at 1 million cells per well. After 24 h, these cells were transfected with 1 μg plasmid using 0.017 mg /ml of Polyethylenimine (PEI) (stock of 1 mg /ml dissolved in 0.2 M HCl) in serum-free media, followed by supplementing serum after 3 h of transfection and cultured for 24 h before proceeding for cysteine starvation experiment procedure.

## Western blotting

For sodium dodecyl sulfate (SDS)-polyacrylamide gel electrophoresis (PAGE), cells were lysed in a buffer containing smDNAse, 1 mM MgSO₄, 1 × Halt Protease and Phosphatase Inhibitor Cocktail (Thermo Scientific), and 1 × RIPA lysis buffer (EMD Millipore). Lysates were mixed with Laemmli sample buffer and heated for 10 min at 70 °C for protein denaturation. Denatured samples were separated by SDS-PAGE at 140 V for 50 min using Tris-glycine-SDS running buffer.

Proteins were transferred onto methanol-activated Amersham PVDF membranes (Cytiva) using methanol-containing transfer buffer at 100 V for 90 min at 4 °C. Membranes were blocked for 1 h at room temperature in 5% milk powder diluted in PBS-T. Primary antibodies recognizing CUL2 (Abcam, #ab166917, 1:1000), CUL5 (Abcam, #ab184177, 1:1000), CDO1 (Proteintech, #12589-1-AP, 1:500), GAPDH (Cell Signaling Technology, #2118, 1:1000), or vinculin (Abcam, # ab129002, 1:1000) were diluted in blocking solution and incubated with the membranes overnight at 4 °C. Following three washes with PBS-T, membranes were incubated with an HRP-conjugated anti-rabbit secondary antibody (Cell Signaling Technology, #7074, 1:2500). After three additional 10-minute washes in PBS-T, membranes were developed using Amersham ECL Prime Western Blotting Detection Reagent (Cytiva) and imaged with the Amersham ImageQuant 800.

## Ubiquitylation assays

Cy5-CDO1 (0.25 μM), CUL2-NEDD8-RBX1 (0.5 μM), LRRC58-EloB/C (0.5 μM), ARIH1 (0.4 μM), UBE2L3 (2.0 μM), UBE2R2 (2.0 μM), and ubiquitin (100 μM) were mixed with buffer (50 mM HEPES pH 7.5, 100 mM NaCl, 7.5 mM MgCl₂, 5 mM ATP, 0.5 mg /mL BSA). For assays done with CUL5, components were mixed as previously described, but CUL5-NEDD8-RBX2 (0.5 μM) and ARIH2 (0.4 μM) were used instead. For the assays comparing ubiquitylation of CDO1 variants by the VHL-Cmpd8 degrader, components were mixed as previously described, but with either Cy5-CDO1 or Cy5-K8R-CDO1 (0.25 μM), either LRRC58-EloB/C or VHL-EloB/C (0.5 μM), either WT-ubiquitin or $K_0$-ubiquitin (100 μM), and either DMSO or Cmpd8 (0.125 μM, NVS-VHL720-MedChemExpress). All reactions were initiated by adding UBA1 (0.1 μM) and proceeded at room temperature with samples taken at time points taken at 0 min, 1 min, 5 min, or 10 min. Samples were quenched through mixing with 3X SDS-PAGE buffer (150 mM Tris-HCl, 20 vol% glycerol, 30 mM EDTA, 4% SDS) before running on hand-cast 4-22% or 12% (SERVA) SDS-PAGE gels. Pulse-chase assays were performed to examine CDO1 ubiquitylation by different cullins and their partner ubiquitylating enzymes. $K_0$-ubiquitin (6.25 μM), UBA1 (0.3 μM), and either UBE2D3, UBE2L3, or UBE2R2 (5 μM) were incubated in 25 mM HEPES pH 7.5, 100 mM NaCl, 100 mM MgCl₂ and 2 mM ATP at room temperature for 30 min. This pulse reaction was quenched by a 1:4 dilution in 25 mM HEPES pH 7.5, 100 mM NaCl, 0.1 mg /mL BSA, and 5 U/mL Apyrase (NEB) and incubated on ice for 5 min. Chase reactions were initiated by adding the quenched pulse reaction, in a 1:1 dilution (final 0.5 μM concentration of E2-Ub), to Cy5-CDO1 (0.5 μM), CUL2-NEDD8-RBX1 or CUL5-NEDD8-RBX2 (0.5 μM), LRRC58-EloB/C (0.5 μM), and either ARIH1 (with CUL2-NEDD8-RBX1) or ARIH2 (with CUL5-NEDD8-RBX2) (0.4 μM), or no additional enzyme, in 25 mM HEPES pH 7.5, 100 mM NaCl, 0.1 mg /mL BSA. A 0 time point was taken before initiation, then the reactions proceeded at room temperature with samples taken at time points of 0 s, 10 s, 20 s, and 1 min. Samples were quenched through mixing with 3X SDS-PAGE buffer (150 mM Tris-HCl, 20 vol% glycerol, 30 mM EDTA, 4% SDS) before running on hand-cast 4–22% or 12% (SERVA) SDS-PAGE gels. Pulse-chase assays to visualize substrate receptor autoubiquitylation by UBE2L3 were performed as above using the same concentration but without the addition of Cy5-CDO1 and using Fluorescein-$K_0$-ubiquitin. LRCC58 was pre-incubated with a threefold excess of CUL2/5-NTD for 10 min where applicable. Autoubiquitylation reactions were performed on ice with samples taken at time points of 0 s, 10 s, and 60 s before running on hand-cast

4-22% or 12% (SERVA) SDS-PAGE gels. Gels were visualized on an Amersham Typhoon Imager (Cytiva).

## Cryo-EM structure determination

**Activity-based probe formation and trapped complex formation.** To visualize ubiquitin transfer to CDO1 an activity-based-probe was employed. Ub-MESNa and Ub-BmDPA were prepared as described previously[44]. 75 μM UB-BmDPA was reacted with 1.2-fold excess of a mutant version of CDO1 (K8C, C76A, C93S, C130A, C164S) in reaction buffer containing 50 mM HEPES pH 8, 150 mM NaCl at 30 °C for 1 h. Subsequently, His-Pulldown was performed to remove unreacted CDO1. Final probe was purified using SEC in 25 mM HEPES, pH 7.4, 150 mM NaCl, 1 mM TCEP. For trapping of ubiquitylation mimics, open variants of ARIH1/ARIH2 were incubated with 5 mM DTT for 10 min on ice. Desalting using Zeba Micro Spin Desalting colums was performed immediately prior of initiating trap formation. In the final trap reaction, neddylated CUL2/CUL5 and LRRC58-EloB/C were at 7.5 μM and UB-CDO1-probe at 30 μM. ARIH1/ARIH2 was added last at a final concentration of 10 μM. The reaction was performed at 30 °C for 1 h and subsequently purified using SEC with a Superose 6 Increase 10/300 GL column. SEC buffer for CUL2-ARIH1 complex consisted of 25 mM HEPES pH 7.4, 100 mM NaCl and 1 mM TCEP, while buffer used for the CUL5-ARIH2 complex was 25 mM HEPES pH 7.4, 150 mM NaCl and 1 mM TCEP. Peak fractions containing a trapped complex were verified by SDS-PAGE, concentrated and subsequently used for cryo-EM sample preparation.

**Cryo-EM sample preparation and collection.** Quantifoil holey-carbon cryo-EM grids R1.2/1.3 200 mesh, Cu mesh grids were glow-discharged, and 4 μL of freshly co-sized and concentrated LRRC58-CDO1-CUL2 complexes were applied to the grids at 3 μM. A Vitrobot Mark IV (Thermo Fisher Scientific) was used to blot samples with a blot force of 2 and a blot time of 2 s at 95% humidity at 4 °C before plunge-freezing into liquid ethane. LRRC58-CDO1-CUL5 complexes were prepared in the same way, but at a concentration of 8 μM with the subsequent addition of 0.05% β-OG and using a blot force of 3 and a blot time of 3 s.

Grids were screened using SerialEM (4.2.1) on a Glacios Cryo-Transmission Electron Microscope operating at 200 kV with a Gatan Alpine Vista Camera to select grids suitable for high-resolution data collection. Screening datasets for LRRC58-CDO1-neddylated CUL2-RBX1-ARIH1-Ub complex were collected at a magnification of x22,000, corresponding to a pixel size of 1.871 Å/px, in a defocus range of -1 to -2.6 μm and with a dose of 60 e⁻/Å² fractionated over 40 frames.

High-resolution datasets were collected on a Titan Krios Cryo-Transmission Electron Microscope operating at 300 kV equipped with a K3 direct electron detector (Gatan) in counting mode and a Bio Quantum post-column energy filter (Gatan). Micrographs were collected at a magnification of x105,000 and pixel size of 0.8512 Å/px. A defocus range between -0.6 and -2.2 μm was chosen, and 30 frames were collected with a total electron dose of ~ 50 e⁻/Å². Data collection was done using SerialEM (4.2.1). Detailed data collection, refinement, and validation statistics are summarized in Table 1.

**Cryo-EM data processing.** For all datasets, processing was done using CryoSPARC (v4.7.1 and v4.7.1 + 250814 (Patch))[95–97] unless otherwise stated.

For the dataset of LRRC58-EloB/C-CDO1 in complex with CUL2-RBX1, first patch motion correction and patch CTF estimation were performed. Particles obtained through blob picking were used in multiple rounds of 2D classification before template picking. Particles obtained through template picking were used in parallel 2D classifications of random particle sets of ~1 million particles. The best classes from these were combined and used to train a Topaz model for particle picking. These particles were used in multiple rounds of 2D

classification, multiclass ab initio reconstruction, heterogeneous refinement, and non-uniform refinement, before a final 4.5 Å volume was obtained, as detailed in Supplementary Fig. 7.

For the datasets of the sample representing CDO1 ubiquitylation by LRRC58 with neddylated CUL2-RBX1-ARIH, first patch motion correction and patch CTF estimation were performed. For the screening dataset, particles obtained through blob picking were extracted (4x binned, box size 108 px, for all datasets) and used in rounds of 2D classification, heterogenous refinement, non-uniform refinement, and 3D classification, before a final 5 Å volume was obtained, as detailed in Supplementary Fig. 9. This volume was used for template picking in the higher-resolution dataset of the same sample, as detailed in Supplementary Fig. 10. After template picking, particles were extracted (4x binned) and used in multiple rounds of heterogenous refinements, before re-extraction at full box size (432 px, for all datasets), further rounds of heterogenous refinement and local refinement to obtain a consensus reconstruction at a 3.73 Å resolution. To obtain higher resolution in areas of interest, local refinements were performed, yielding locally refined maps at resolutions ranging from 3.54–3.67 Å. Locally refined maps were combined with the consensus map to create a composite map at 3.7 Å resolution using Frankenmap in Warp (1.9.0)[98]. This reconstruction showed considerable orientation bias and poor resolution in multiple areas that were not alleviated by local refinements, as exhibited in Supplementary Fig. 10, thereby preventing accurate atomic model building. Further 2D classification, 3D classification, and local refinement steps were explored, but did not yield an improved reconstruction. The CUL5-associated complex did not suffer from as much orientation preference and was able to be used to unambiguously build a model.

For the dataset of LRRC58-EloB/C-CDO1 in complex with CUL5-RBX2-ARIH2, first patch motion correction and patch CTF estimation were performed. Particles obtained through blob picking were used in 2D classification before ab initio reconstruction. Both a good and a poor ab initio reconstruction were used as templates in 2 rounds of heterogeneous refinement. The best particles were used in non-uniform refinement to obtain a final 5.39 Å volume, as detailed in Supplementary Fig. 8.

For the datasets of the sample representing CDO1 ubiquitylation by LRRC58 with neddylated CUL5-RBX2-ARIH2, patch motion correction and patch CTF estimation were performed before template picking with a volume obtained from the CryoSPARC Live[95] processing session of the same dataset. Particles were extracted (4x binned) before multiple rounds of heterogeneous and non-uniform refinement, as detailed in Supplementary Fig. 11, to yield a 2.91 Å consensus reconstruction. Particle subtraction and local refinements were performed on the substrate binding areas to result in a 3.05 Å map, which was combined with the consensus reconstruction using Frankenmap in Warp (1.9.0) to create a 2.95 Å composite map[98].

**Model building and refinement.** Maps were sharpened using DeepEMhancer (version 2020.09.07) or sharpening in CryoSPARC with a manually set B-factor of -60.

As the reconstruction resolution was too poor to unambiguously build a structural model for all parts of the sample representing CDO1 ubiquitylation by LRRC58 with neddylated CUL2-RBX1-ARIH1, structure coordinates were rigid-body docked piecewise, as described below, into density in UCSF ChimeraX (1.9) then combined as a singular, multichain model; EloB/C-CUL2 N-terminal domain (EloB residues 1–81, 88–97, EloC residues 8–111, CUL2 N-terminal domain residues 1–116, 135–275, from PDB: 8WQH), CUL2 C-terminal domain (residues 276–381, 384–496, 503–626, 633–647, from PDB: 8Q7R), ARIH1-RBX1 (ARIH1 residues 420–554, RBX1 residues 21–59, 67–108, from PDB: 7B5M), CDO1-LRRC58 portions built in the structure of LRRC58-EloB/C-CDO1 in complex with neddylated CUL5-RBX2-

ARIH2-Ub, and AlphaFold3 model of ARIH1-Ub (ARIH1 residues 316–418, and Ub residues 1–76).

An initial structure of the sample representing CDO1 ubiquitylation by LRRC58 with neddylated CUL5-RBX2-ARIH2 was built by rigid body fitting the following coordinates into density in UCSF ChimeraX (1.9); for CUL5 (residues 12–266) and EloB/C the previously published structure 4N9F was used. CUL5 (residues 267–768), NEDD8, RBX2 and ARIH2 were taken from 9SDX. Ubiquitin was fit using 7B5M. For LRRC58 and CDO1, an initial model was generated using AlphaFold3[99], including the binding partners EloB/C. In addition, an AlphaFold3 model of Ub-ARIH2 (residues 283–356) was generated and fit onto the ubiquitin to improve modeling around the active site of ARIH2. This initial model was iteratively improved by manual building and fitting in Coot[100] and real-space refinement in PHENIX[101]. Structural figures were created using UCSF ChimeraX (1.9).

The chimeric structural model of VHL-Cmpd8-CDO1 in complex with CUL2$^{NTD}$-EloB/C used in Fig. 6d was made as follows: The VHL portions of the EloB/C-VHL-CUL2$^{NTD}$ crystal structure (PDB: 4WQO) and the crystal structure of EloB/C-VHL-CDO1 complex bound to Cmpd8 (PDB: 8VL9) were aligned in UCSF ChimeraX (1.9), with an RSMD of 0.638 Å. Then the CUL2$^{NTD}$ portion of 4WQO was grafted onto the rest of the 8VL9 structure to create the chimeric structural model.

## Statistical analyses

Statistical analyses performed for individual data are described above. In summary, bars represent the average values for the specified number of replicates, with all error bars reporting the standard deviation of the data points. Means and standard deviations were calculated using GraphPad Prism (10.4.2) and GraphPad Prism (10.6.1). For proteomics data, the Perseus software package (2.0.9.0) was used to perform $t$ tests (two-sided, permutation-based FDR calculation, 0.05 FDR, 250 randomizations, $s0 = 0.1$) and generate volcano plots with 5% FDR threshold curves. For RNA-Seq data, differentially expressed genes were identified using the DESeq2 package in the R statistical environment (4.3.2), and a stringent threshold of an adjusted $p$-value (Benjamini-Hochberg procedure) $\leq 0.01$ was applied to define statistically significant differentially expressed genes.

## Reporting summary

Further information on research design is available in the Nature Portfolio Reporting Summary linked to this article.

## Data availability

The structure data for the sample representing CDO1 ubiquitylation by LRRC58 with neddylated CUL5-RBX2-ARIH2 are available from the RCSB-PDB and EMDB with the identifiers 9T7V, EMD-55658, EMD-55659, EMD-55660. The cryo-EM volumes for the sample representing CDO1 ubiquitylation by LRRC58 with neddylated CUL2-RBX1-ARIH1 are available from the EMDB with the identifiers EMD-55652, EMD-55653, EMD-55654, EMD-55655, EMD-55656. The mass spectrometry proteomics data have been deposited to the ProteomeXchange Consortium via the PRIDE partner repository with the identifier; PXD071631. The RNA-Seq data have been deposited in NCBI's Gene Expression Omnibus and are accessible through GEO Series accession number GSE309930.

Supplementary Data 1 contains the mass spectrometry data tables. Raw gel and blot images are provided as source data. Data points represented in volcano plots, bar charts, and other plots are provided as source data. Source data are provided in this paper.

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

## Acknowledgements

We thank Charlotte Duteil for advice on flow cytometry, Jakob Farnung for advice on assay design, Sara Šepić and Rajan Prabu for advice on cryo-EM processing, Joana Cobaj for sharing of reagents and Josef Kellermann and all other members of the Schulman lab for general advice and assistance. We also thank Barbara Steigenberger and Victoria Sánchez at the MPIB Mass Spectrometry Facility (RRID:SCR_025745), Rinho Kim at the MPIB NGS Core Facility (RRID:SCR_025746), Assa Yeroslaviz at the Bioinformatics Core Facility (RRID:SCR_025742), Martin Spitaler and Markus Oster at the MPIB Imaging Facility (RRID:SCR_025739), Stephan Uebel and Stefan Pettera at the MPIB Biochemistry Core Facility (RRID:SCR_025743) and Daniel Bollschweiler and Tillman Schäfer at the MPIB Cryo-EM Facility (RRID:SCR_025744). L.J.S, A.T., and S.A.M. disclose support for this research from PhD fellowships from the Boehringer Ingelheim Fonds. G.K. discloses support for this research from the NIH, R01GM141409 and R01CA279255. M.M., P.J.M, and B.A.S. disclose support for this research from the Max Planck Society. B.A.S. discloses support for this research from the European Union (ERC, UPSmeetMet, 101098161, B.A.S.). Views and opinions expressed are, however, those of the authors only and do not necessarily reflect those of the European Union or the European Research Council. Neither the European Union nor the granting authority can be held responsible for them. G.A.A., K.S., L.S., K.V.G., J.D., S.vG., C.S., J.M., L.T.H., and C.G. declare no relevant funding.

## Author contributions

G.A.A. and L.J.S. contributed equally to this work, and they both have the right to be listed first in bibliographic documents. Conception: P.J.M., B.A.S., G.A.A., L.J.S., A.S.T., and L.T.H.; Biochemistry: G.A.A. and L.J.S.; Cryo-EM, structure building and refinement: G.A.A. and L.J.S.; Proteomics: G.A.A. and L.T.H.; Cell biology: K.S., A.S.T., P.J.M., G.A.A., C.G., J.M., and K.V.G; Reagent production: G.A.A., L.J.S., L. S., S.A.M., J.D., S.v.G., and C.S.; Data analysis: G.A.A., L.J.S., K.S., A.S.T., L.T.H., K.V.G, B.A.S., and P.J.M.; Supervision: B.A.S., P.J.M., M.M., and G.K.; Paper preparation: G.A.A., L.J.S., G.K., P.J.M., and B.A.S., with input from all authors.

## Funding

## Competing interests

B.A.S. is a member of the scientific advisory boards of Proxygen and Lyterian. The other authors declare no competing interests.
