## [Transparent Peer Review file · Nature Communications]

Cysteine availability tunes ubiquitin signaling via inverse stability of LRRC58 E3 ligase and its substrate CDO1

Corresponding Author: Professor Brenda A. Schulman

Version 0:

Reviewer comments:

Reviewer #1

(Remarks to the Author)

This manuscript by Andree et al. describes how the ubiquitin-proteasome system regulates metabolism in response to the abundance of cysteine, uncovering the LRRC58 E3 ligase and its substrate the metabolic enzyme cysteine dioxygenase (CDO1). Proteomics approaches are elegantly combined with biochemistry and structural biology. Overall this is a wonderful manuscript that independently confirms recently published findings as properly cited (Xiao, H. et al. Covariation MS uncovers a protein that controls cysteine catabolism. *Nature* (2025). <https://doi.org/10.1038/s41586-025-09535-5>) and (Ramage, D. E. et al. LRRC58 defines an E3 ubiquitin ligase complex sensitive to cysteine abundance. *bioRxiv*, 2025.2009.2023.678073 (2025). <https://doi.org/10.1101/2025.09.23.678073>). Critically, this manuscript extensively employs state-of-the-art cryo-EM for structure analysis, whereas Xiao et al. and Ramage et al. employ AlphaFold modelling. This is a vital contribution of the current manuscript compared to the published manuscript by Xiao et al. and the preprint by Ramage et al. This manuscript would obviously benefit from rapid publication.

I have a few minor comments:

1. Figures 1 and 2 would benefit from a few immunoblots to independently confirm the mass spec identification of LRRC58 and CDO1.
2. Figure 1d would benefit from a more distinguishing color palette.
3. Figure 3c: it would be useful to confirm efficient LRRC58 knockdown by immunoblotting.
4. Figure 4: it would be useful to include the identity of molecular glue degrader Compound 8.

Reviewer #2

(Remarks to the Author)

Andree et al. describe their work structurally characterising CUL2/CUL5-LRRC58 as the native E3 ligase responsible for CDO1 degradation and compare its ubiquitination mechanism to CUL2-VHL used in TPD strategies. This work builds on recent publications that independently identified LRRC58 as a cysteine-mediated CDO1 degrader, and the authors extend this work by comparing native and molecular glue-induced recognition mechanisms. The cryo-EM structures of the CUL2-LRRC58 and CUL5-LRRC58 are new additions to the field and, in principle, offer an opportunity to rationalise cullin selection, substrate positioning, and lysine specificity. However, several aspects of the study require clarification or further analysis. In particular, the physiological relevance of CUL2 versus CUL5 remains ambiguous, the interpretation of structures derived from chemically stabilised ubiquitination intermediates is not sufficiently contextualised, and key mechanistic claims, such as Lys8 selectivity, LRRC58 stabilisation under cysteine starvation, and the robustness of molecular glue-mediated ubiquitination, are not fully supported by comparative structural or biochemical rationale. Additionally, the structural descriptions and figure presentations are at times difficult to follow, and a more thorough integration of the new structures with existing literature would substantially strengthen the manuscript.

Major points

1. The finding that LRR58 interacts with EoBC and can recruit either CUL2 or CUL5 is interesting and aligns with the findings of Ramage et al. (2025, bioRxiv). The rationale for focusing primarily on CUL2 is not clearly articulated. The siRNA knockdown data suggest that CUL2 depletion has a stronger effect than CUL5 depletion, but this should be explicitly discussed. Additionally, the structural analysis is performed with CUL5 while the biochemical ubiquitination assays use CUL2. This dual usage is currently ambiguous. The authors should clarify whether CUL5 contributes under specific conditions or whether CUL2 is the predominant physiological partner. In particular, the mutagenesis and in vitro ubiquitylation assays in Figure 4 only test CUL2-LRR58 and not CUL5-LRR58. This should be repeated for CUL5-LRR58 to test for any cullin specific differences.

2. The cryo-EM structures in this study were determined using chemically stabilised ubiquitination intermediates incorporating ARIH-family enzymes, which effectively trap a single catalytic state. While this approach is well established and appropriate for structural analysis, it does not necessarily capture the full physiological ubiquitination space or consider potential alternative conformational heterogeneity. Thus, while the observed geometry may represent a catalytic configuration underlying Lys8 specificity, it may not encompass the full range of transient states sampled during CDO1 ubiquitination in vivo. The authors should therefore provide a more detailed justification for their choice of the reconstituted system. In particular, statements such as “Previous studies have shown that ARIH-family enzymes broadly and efficiently target CRL substrates with a wide range of structural features” (Lines 269-280) are overly general and would benefit from clearer explanation and supporting detail.

3. The authors use RNA sequencing analysis in HEK293T cells to show that LRR58 mRNA levels do not change upon cysteine starvation, suggesting post-transcriptional regulation. Using proteomics, they go on to show that LRR58 is detectable primarily under cysteine-free conditions but accumulates in normal media in the presence of proteasome or neddylation inhibitors, suggesting CRL involvement. The suggestion that LRR58 is continuously degraded under normal conditions but stabilised during cysteine starvation is valid but remains speculative. A direct in vitro auto-ubiquitination assay or experiments using LRR58 mutants defective in EoBC or cullin binding would strengthen this claim. The authors could also reference prior work on BC box receptor autodegradation (e.g., Ramage et al. 2025) to contextualise the hypothesis.

4. A focus of this manuscript is the observation that VHL-compound-8 retains binding and catalytic activity even when mutations perturb the LRR58-CDO1 interface, enabling ubiquitination of different lysines. Since the structure of the VHL-CDO1-compound-8 complex has been solved (PDB 8VL9), it would strengthen the manuscript to compare the binding interfaces directly. The current Figure 6 illustrates the D-patch interface but does not reveal differences in positioning, binding mode, or lysines in proximity to the ubiquitination machinery. A figure showing VHL-compound-8-CDO1 aligned with a CUL2-VHL structure, followed by comparison to the LRR58 complex, highlighting relative orientations and nearby lysines, would greatly help the reader. Providing structural rationale for why particular mutations are disruptive (or not) would also be beneficial.

5. Similarly, the authors report that compound-8-mediated ubiquitination via VHL is more efficient than LRR58-mediated ubiquitination in vitro. It would be valuable to note that this likely reflects the higher affinity of the molecular glue complex for CDO1 relative to LRR58 (0.1 M Kd for VHL-compound8-CDO1 and 4.67 M Kd for LRR58 and CDO1). It is important to clarify this and make the distinction between increased ternary complex formation and more efficient ubiquitin transfer.

6. The manuscript only briefly mentions the structural differences between the CUL2-LRR58 and CUL5-LRR58 complexes but does not provide sufficient context or interpretation of what these differences might mean functionally. A more detailed comparison, including potential implications for substrate geometry or ubiquitination site selection, is needed. The structural portion of the manuscript as a whole is difficult to follow and would benefit from clearer explanation. This paragraph should be revised for clarity and readability.

7. The authors describe extra density between LRR9, the C cap, and CDO1 as “resembling a molecular glue.” This is a somewhat curious analogy, given that the authors then proceed to model the final four LRR58 residues into this region based on AlphaFold predictions. The rationale for this interpretation should be clarified. In Extended Data Figure 9, these four LRR58 residues are modelled into the additional density; however, the quality of the density and the extent to which the model is supported by the map are difficult to assess. In addition, the authors do not discuss any potential interactions between these residues and CDO1, which is surprising given the proposed functional relevance of this motif. To strengthen this interpretation, the authors should report Q-scores for these residues, provide a close-up view of the corresponding map region and identify any putative contacts with CDO1 that could explain why the C-terminal tail of LRR58 is ordered. Given the apparent functional importance of this motif, mutagenesis of these LRR58 residues, especially in light of the recent bioRxiv study by Ramage et al., 2025 highlighting the motif as essential for cysteine-mediated instability, would provide a direct test of their contribution to CDO1 binding.

Minor points

8. Figure 1d and associated text (lines 138-189). The labelling in Figure 1d is unclear. Improve colour use and match the key order to the bar graph. The statement that “LRR58 amounts change to a striking extent” is not adequately supported by the data as presented. The authors do not quantify fold-changes or statistical confidence, or comment on baseline LRR58 expression differences across cell types.

9. Figure 5. The colour schemes for RBX1/2 and ARIH1/2 are too similar, making the models difficult to distinguish. More distinct colours are recommended. Panels 5b, 5c, 5d, and 5e are mislabelled in the legend and should be corrected. In Figure 5e, the zinc ion is not visually prominent; a more contrasting colour would improve clarity.
10. Lines 303-304: The description of the “C cap” and “several LRRs” forming the substrate binding domain is too vague. The authors should specify which LRRs contribute to substrate engagement.
11. Lines 323-325: The authors state that the LRRC58 C terminus could not be fully resolved. Please clarify which residues lack density, and it would be helpful if the disordered regions of LRRC58 were highlighted in the LRRC58 schematic of Figure 5a.
12. Figure 6a-c. Please label the proteins in these panels, as is done in Figure 5.
13. Line 333. The sentence should read: “The ‘top side’ of CDO1 engages a continuous surface with LRRC58...” rather than “in LRRC58.”
14. Line 245. The authors state that “both cellular and in vitro ubiquitylation assays” were performed, but only a degradation reporter assay is shown for cellular experiments. The authors should clarify this.
15. Figure 4a. The presentation is confusing for the reader as one is comparing +/- cysteine and the other is +/- compound. Clearer labelling would help.
16. Extended Figure 5 and 6 would benefit from 2D classes being shown similar to Extended Figure 4. Inclusion of local resolution in each figure would also be useful.
17. The authors describe the limited resolution of CUL2-LRRC58 as due to preferred orientation. However, in Extended Figure 5 the orientation distribution plot shows moderate preferred orientation. Authors should comment on any heterogeneity and continuous flexibility within the sample that could be contributing to the lower resolution.

Reviewer #3

(Remarks to the Author)

This study investigates LRRC58 as a cysteine-responsive BC-box substrate receptor that targets the metabolic enzyme CDO1 for ubiquitin-mediated degradation. Using a combination of DIA-MS, reporter assays, biochemical reconstitution, and high-resolution cryo-EM of trapped ubiquitylation intermediates, the authors uncover how LRRC58 selectively positions CDO1 for ubiquitylation at Lys8. The structures reveal a previously undescribed LRRC58 architecture with a C-terminal cap domain that engages multiple interaction patches on CDO1. Comparative analyses of CUL2- versus CUL5-assembled CRLs highlight conserved catalytic organization and provide mechanistic insight into selective substrate targeting. Overall, this is a comprehensive and technically impressive study. However, several conceptual and interpretational issues, particularly in the early Results sections, limit the strength of some conclusions, where correlation is occasionally interpreted as functional activation.

Major concerns

1. Throughout the manuscript, the authors state that the LRRC58–CRL is “activated” upon cysteine starvation. While the authors convincingly demonstrate CRL dependence using MLN4924 and show increased association of LRRC58 with neddylated cullins and elevated LRRC58 protein abundance under cysteine-starved conditions, these data establish requirement rather than increased enzymatic activity. Enrichment in an activity-based CRL pulldown does not distinguish between increased protein abundance, altered binding stability, reduced deneddylation, or true catalytic activation. Although Figure 3 convincingly demonstrates LRRC58-dependent ubiquitination of CDO1 and CRL dependence, these experiments establish biochemical capability rather than cysteine-dependent activation of the LRRC58–CRL. Direct measurements of increased ubiquitination activity under cysteine starvation are not provided, and the language should therefore be tempered accordingly.
2. The authors provide convincing genetic evidence that CUL2 is the dominant cullin mediating CDO1 degradation in cells, with CUL5 contributing redundantly, and they further demonstrate that LRRC58 can assemble functional CRLs with both cullins in vitro. However, the active-CRL Fab used for profiling recognizes neddylated CUL1, CUL2, CUL3, and CUL4, but not CUL5, meaning that LRRC58–CUL5 assemblies are not detectable in the initial discovery experiments. This limitation is not explicitly discussed in the Results section and creates a narrative gap between the early profiling data and the later structural emphasis on CUL5-based complexes. Clarifying whether CUL5 engagement is inferred, context-dependent, or primarily supported by biochemical and structural feasibility would strengthen the coherence of the study.
3. The authors present convincing evidence that LRRC58 is post-transcriptionally regulated in a proteasome- and CRL-dependent manner, as shown by stabilization of LRRC58 protein upon MG132 and MLN4924 treatment together with unchanged LRRC58 mRNA levels. However, some of the phrasing suggests continuous or constitutive degradation of LRRC58 under cysteine-replete conditions, which is not directly measured. Slightly tempering this language would better

align the conclusions with the data presented.

4. The manuscript presents strong convergent evidence linking LRR58 to CDO1 degradation through proteomics, genetic dependency, cell-based reporters, biochemical reconstitution, and parallel literature. However, no endogenous LRR58–CDO1 interaction is demonstrated in cells, and all physical interaction evidence derives from in vitro assays or computational predictions. While this may reflect technical limitations, clarifying the predictive nature of these interactions or explicitly acknowledging this limitation would strengthen the in vivo interpretation.

5. The authors convincingly demonstrate that LRR58-mediated degradation of CDO1 is highly selective for Lys8, whereas compound-8-mediated degradation is lysine-promiscuous. However, the manuscript further concludes that the reduced efficiency of the native pathway relative to degrader-induced degradation primarily results from this lysine selectivity. While this is a plausible explanation, alternative factors such as differences in E2 recruitment, catalytic geometry, initiation versus elongation kinetics, or chain architecture are not examined. Thus, lysine selectivity is clearly supported, but its role as the primary determinant of degradation efficiency is inferred rather than directly demonstrated.

6. Several structural interpretations, particularly for the CUL2-based complex, appear to extend beyond what is directly supported by map quality. It would be helpful if the authors more explicitly distinguished regions supported by cryo-EM density from those incorporated via rigid-body docking or AlphaFold predictions. Given that the positioning of Lys8 within approximately 25 Å of the ARIH active site is central to the proposed mechanism, additional information on local resolution and density quality in this region would strengthen confidence in this conclusion. Furthermore, although in vivo data support CUL2 as the dominant cullin for CDO1 turnover, the functional relevance of structural differences between CUL2- and CUL5-based complexes remains unclear and would benefit from further discussion.

7. Finally, given the existence of a parallel study independently identifying the LRR58–CDO1 pathway, the authors should ensure that claims of novelty throughout the Introduction and Discussion clearly emphasize the unique mechanistic and structural insights provided here, rather than the initial discovery of the pathway itself. In particular, Ramage et al. report LRR58-dependent regulation of CDO1 stability and cysteine homeostasis in vivo, including metabolic and physiological consequences, which places the present work in the context of a parallel pathway discovery and highlights its primary strength in structural and mechanistic resolution rather than pathway identification.

Minor concerns

1. Proteomics figures should more clearly distinguish true absence from below-detection abundance.
2. Local resolution overlays at the LRR58–CDO1 interface would improve interpretability.
3. Composite figures would benefit from clearer labeling of experimental density versus fitted models.
4. Terminology for LRR58 C-terminal regions, for example “C-cap”, should be standardized.
5. A schematic defining CDO1 interface patches and the D-patch would aid clarity.
6. Brief discussion of potential cooperativity among interface patches would be useful.
7. Additional contextualization of ARIH1 and ARIH2 architecture relative to prior work would help orient non-specialist readers.
8. In Figure 4b, the time axis annotation in minutes is missing at the indicated position and should be added for consistency with the right panel.

Reviewer #4

(Remarks to the Author)

Version 1:

Reviewer comments:

Reviewer #2

(Remarks to the Author)

the authors have addressed all concerns sufficiently well and clarified all points raised. Happy to recommend it for publication.

Reviewer #3

(Remarks to the Author)

The authors have carefully revised the manuscript and addressed the concerns raised in the initial review. The revised version includes additional experiments, particularly the LRR58 knockout analyses and expanded biochemical assays, which strengthen the mechanistic link between LRR58 and CDO1 regulation. The authors have also clarified the interpretation of CUL2 versus CUL5 usage, improved the presentation and explanation of the structural data, and appropriately moderated statements that previously implied catalytic activation of the LRR58–CRL complex under cysteine

starvation.

The structural analysis is now presented more clearly, with improved distinction between cryo-EM density supported regions and modeled components, and the inclusion of local resolution information strengthens confidence in the structural interpretations. The manuscript also better contextualizes its findings relative to recent studies describing the LRRC58–CDO1 pathway.

Overall, the revisions satisfactorily address the major conceptual and technical concerns raised in the previous review. I have no further major concerns and believe the manuscript is suitable for publication.

Reviewer #4

(Remarks to the Author)

Responses to Reviews

“Cysteine availability tunes ubiquitin signaling via inverse stability of LRRC58 E3 ligase and its substrate CDO1”

We were very pleased by the Reviewer’s enthusiasm for our study! We also appreciated the suggestions to improve our study. Addressing the Reviewer comments improved the presentation, clarity, and quality of the manuscript. Changes to the text and legends that address reviews are highlighted in cyan in the revised manuscript.

While performing experiments to address the direct requests from reviewers, we were able to obtain additional data to further strengthen our study. Most notably, we have now generated an LRRC58 knockout cell line, which provides additional evidence for regulation of CDO1 by LRRC58 at an endogenous level. Furthermore, we added experiments with a few additional CDO1 mutants. We realized that our original “Patch-4” mutant used to compare ubiquitylation via LRRC58 versus the Compound8-VHL E3s did not include a key interacting residue, and thus did not eliminate the crucial contacts with LRRC58. For the revised, we have replaced the original with a new Patch-4 mutant that is more comprehensive, and indeed this confirms a role for Patch-4 in ubiquitylation. Second, our original mutational scan included Ala substitutions at the sites of disease mutations mapping to the CDO1 interface with LRRC58. We now have added the actual reported disease substitutions to our mutational screen. These data substantiate our original findings showing (1) distinct endogenous versus degrader regulation of CDO1, and (2) that a molecular degrader can rescue the defects in ubiquitylation caused by CDO1 disease mutants mapping to the interface with CDO1.

In addition, in response to Reviewer comments we made a small change to the LRRC58-CDO1-CUL5 structure (removing a few residues, at the C-terminus of LRRC58). We thus now provide the revised file to the reviewers at the original data link provided again here: [editorial note: temporary access link redacted as now redundant]

REVIEWER COMMENTS

Reviewer #1 (Remarks to the Author):

This manuscript by Andree et al. describes how the ubiquitin-proteasome system regulates metabolism in response to the abundance of cysteine, uncovering the LRRC58 E3 ligase and its substrate the metabolic enzyme cysteine dioxygenase (CDO1). Proteomics approaches are elegantly combined with biochemistry and structural biology. Overall this is a wonderful manuscript that independently confirms recently published findings as properly cited (Xiao, H. et al. Covariation MS uncovers a protein that controls cysteine catabolism. *Nature* (2025). <https://doi.org:10.1038/s41586-025-09535-5>) and (Ramage, D. E. et al. LRRC58 defines an E3 ubiquitin ligase complex sensitive to cysteine abundance. *bioRxiv*, 2025.2009.2023.678073 (2025). <https://doi.org:10.1101/2025.09.23.678073>). Critically, this manuscript extensively employs state-of-the-art cryo-EM for structure analysis, whereas Xiao et al. and Ramage et al. employ AlphaFold modelling. This is a vital contribution of the current manuscript compared to the published manuscript by Xiao et al. and the preprint by Ramage et al. This manuscript would obviously benefit from rapid publication.

We are pleased by the Reviewer's enthusiasm for our work.

I have a few minor comments:

1. Figures 1 and 2 would benefit from a few immunoblots to independently confirm the mass spec identification of LRRC58 and CDO1.

Unfortunately, there is presently no functional antibody for LRRC58. The complementary studies (Xiao et al. Nature; Ramage et al., bioRxiv) also were unable to find a commercial antibody that recognizes LRRC58. We tried to generate antibodies with two different antigens, but unfortunately those efforts were not successful. As an alternative approach to address this point and add validity to the initial mass spectrometry identification, we have now generated a HEK293T knockout (KO) cell line of LRRC58. Results with the knockout mirror our original findings: endogenous CDO1 levels increase in the absence of LRRC58, and decrease when a rescue vector is introduced. Importantly, unlike WT control cells, LRRC58 is not detected by mass spec in the KO cells even under Cys starvation conditions. See updated Fig. 2f-h.

2. Figure 1d would benefit from a more distinguishing color palette.

Reevaluating the original figure 1d, we realized that what cell line corresponded to each bar would indeed benefit from clarification. We addressed this (and a related comment from Reviewer #2) in the revised manuscript by labeling each bar with the cell line used.

3. Figure 3c: it would be useful to confirm efficient LRRC58 knockdown by immunoblotting.

As discussed above, we hope that the newly provided knockout cell data equally adequately addresses the raised point in absence of a functioning antibody. See updated Fig. 2f-h.

4. Figure 4: it would be useful to include the identity of molecular glue degrader Compound 8.

Thank you for the suggestion. We have updated Fig. 4a accordingly.

Reviewer #2 (Remarks to the Author):

Andree et al. describe their work structurally characterising CUL2/CUL5-LRRC58 as the native E3 ligase responsible for CDO1 degradation and compare its ubiquitination mechanism to CUL2-VHL used in TPD strategies. This work builds on recent publications that independently identified LRRC58 as a cysteine-mediated CDO1 degrader, and the authors extend this work by comparing native and molecular glue-induced recognition mechanisms. The cryo-EM structures of the CUL2-LRRC58 and CUL5-LRRC58 are new additions to the field and, in principle, offer an opportunity to rationalise cullin selection, substrate positioning, and lysine specificity.

We are pleased by the Reviewer's enthusiasm for our work.

However, several aspects of the study require clarification or further analysis. In particular, the physiological relevance of CUL2 versus CUL5 remains ambiguous, the interpretation of structures derived from chemically stabilised ubiquitination intermediates is not sufficiently contextualised, and key mechanistic claims, such as Lys8 selectivity, LRRC58 stabilisation under cysteine starvation, and the robustness of molecular glue-mediated ubiquitination, are not fully supported by comparative structural or biochemical rationale. Additionally, the structural descriptions and figure presentations are at times difficult to follow, and a more thorough integration of the new structures with existing literature would substantially strengthen the manuscript.

Major points

1. The finding that LRRC58 interacts with EloBC and can recruit either CUL2 or CUL5 is interesting and aligns with the findings of Ramage et al. (2025, bioRxiv). The rationale for focusing primarily on CUL2 is not clearly articulated. The siRNA knockdown data suggest that CUL2 depletion has a stronger effect than CUL5 depletion, but this should be explicitly discussed. Additionally, the structural analysis is performed with CUL5 while the biochemical ubiquitination assays use CUL2. This dual usage is currently ambiguous. The authors should clarify whether CUL5 contributes under specific conditions or whether CUL2 is the predominant physiological partner. In particular, the mutagenesis and in vitro ubiquitylation assays in Figure 4 only test CUL2-LRRC58 and not CUL5-LRRC58. This should be repeated for CUL5-LRRC58 to test for any cullin specific differences.

We apologize for lack of clarity about this issue in our initial manuscript. To address this point, and related comments from Reviewer #3, we now further explain that our activity-based probe only recognizes CUL1, CUL2, CUL3 and CUL4A/B, but that because those experiments do not assay CUL5, which was observed several years ago to bind LRRC58, we tested both CRLs for ubiquitylation activity and found they are both functional. Thus, we sought cryo-EM data for both CUL2 and CUL5, although the map with CUL5 is higher resolution as the sample with CUL2 had greater orientation bias.

We performed several additional experiments to address the Reviewer points. First, we examined effects of CUL2 and CUL5 knockdown on endogenous CDO1 in HEK293T cells. Endogenous CDO1 levels increased in low cysteine only upon simultaneous knockdown of both cullins (Figure 3d). Thus, at least in some settings, the two cullins appear functionally redundant. Second, we performed ubiquitylation assays for the CDO1 mutants with CUL5 as well as CUL2. There were no obvious cullin-specific differences. The new data are shown in Extended Data Figures 1a, 4b, 5c-d, 8c of the revised manuscript. We also added a brief discussion of these findings and work of others on dual CUL2/5 use of LRRC58 and speculate that the different cullins may function in distinct physiological settings.

We also clarify that the knockdown of CUL2, but not of CUL5, was sufficient to stabilize our CDO1 reporter in HEK293T cells. This finding allows extending comparison to the regulation driven by the distinct Compound 8-VHL substrate receptor from the in vitro assays to the reporter stability assays.

2. The cryo-EM structures in this study were determined using chemically stabilised ubiquitination intermediates incorporating ARIH-family enzymes, which effectively trap a single catalytic state. While this approach is well established and appropriate for structural analysis, it does not necessarily capture the full physiological ubiquitination space or consider potential alternative conformational heterogeneity.

Thus, while the observed geometry may represent a catalytic configuration underlying Lys8 specificity, it may not encompass the full range of transient states sampled during CDO1 ubiquitination in vivo. The authors should therefore provide a more detailed justification for their choice of the reconstituted system. In particular, statements such as “Previous studies have shown that ARIH-family enzymes broadly and efficiently target CRL substrates with a wide range of structural features” (Lines 269-280) are overly general and would benefit from clearer explanation and supporting detail.

As cryo-EM structure determination is an empirical process, we attempted to obtain structures with several samples. That said, our goal was to understand how LRRC58 CRLs direct CDO1 ubiquitylation, and thus we were gratified to obtain the best data for the samples representing functional E3s.

To address the reviewer, now include our initial structural work on noncovalent complexes. The portions of the complexes that we could visualize in those samples do show similar assemblies to the ubiquitylation mimics, although poor resolution of the complex with CUL2, and inability to visualize the LRRC58-CDO1 portion of the complex with CUL5 limited understanding from those samples (Extended Data Figure 5a,b).

In the revised manuscript, we also add the data testing CRL partner ubiquitylation enzymes toward CDO1. These experiments show the reactions are more efficient with ARIH-family enzymes, rationalizing their use for the structural analyses (Extended Data Figure 5c,d).

We further added the following sentence regarding the visualized conformation: “Although we cannot rule out that our chemical mimics each capture one of a range of conformations along the ubiquitylation trajectories, they nonetheless show how CDO1 Lys 8 is projected towards the CRL catalytic machineries.”

3. The authors use RNA sequencing analysis in HEK293T cells to show that LRRC58 mRNA levels do not change upon cysteine starvation, suggesting post-transcriptional regulation. Using proteomics, they go on to show that LRRC58 is detectable primarily under cysteine-free conditions but accumulates in normal media in the presence of proteasome or neddylation inhibitors, suggesting CRL involvement. The suggestion that LRRC58 is continuously degraded under normal conditions but stabilised during cysteine starvation is valid but remains speculative. A direct in vitro auto-ubiquitination assay or experiments using LRRC58 mutants defective in EloBC or cullin binding would strengthen this claim. The authors could also reference prior work on BC box receptor autodegradation (e.g., Ramage et al. 2025) to contextualise the hypothesis.

We addressed this in two ways. First, as recommended by the reviewer, we performed autoubiquitylation assays. The data show that both CUL2- and CUL5-based CRL complexes can autoubiquitylate LRR58 (Extended Data Fig 1 a). Second, we performed cullin competition experiments, by pre-incubation of LRR58 with CUL2 or CUL5 N-terminal Domain (NTD) prior to adding the ubiquitylation components of the assays. Formation of LRR58 complexes with the inactive cullin NTDs would competitively inhibit binding to the neddylated, active cullin-RING complexes required for activity. Indeed, adding the cullin NTDs decreased the observed autoubiquitylation.

4. A focus of this manuscript is the observation that VHL-compound-8 retains binding and catalytic activity even when mutations perturb the LRR58-CDO1 interface, enabling ubiquitination of different lysines. Since the structure of the VHL-CDO1-compound-8 complex has been solved (PDB 8VL9), it would strengthen the manuscript to compare the binding interfaces directly. The current Figure 6 illustrates the D-patch interface but does not reveal differences in positioning, binding mode, or lysines in proximity to the ubiquitination machinery. A figure showing VHL-compound-8-CDO1 aligned with a CUL2-VHL structure, followed by comparison to the LRR58 complex, highlighting relative orientations and nearby lysines, would greatly help the reader. Providing structural rationale for why particular mutations are disruptive (or not) would also be beneficial.

We apologize for lack of clarity and we have now included an updated Figure 6d and discussions that highlight differences in CDO1 presentation by the two modes of substrate recruitment. Furthermore, we have added additional rationale to the text in regard to mutational disruptions of CDO1 binding to LRR58 versus the Compound 8-VHL complex.

5. Similarly, the authors report that compound-8-mediated ubiquitination via VHL is more efficient than LRR58-mediated ubiquitination in vitro. It would be valuable to note that this likely reflects the higher affinity of the molecular glue complex for CDO1 relative to LRR58 (0.1 μ M Kd for VHL-compound8-CDO1 and 4.67 μ M Kd for LRR58 and CDO1). It is important to clarify this and make the distinction between increased ternary complex formation and more efficient ubiquitin transfer.

This is an excellent suggestion and we have added this to the text, as follows: “This could result from the relatively higher degrader-induced E3-substrate affinity^{54,55}”.

6. The manuscript only briefly mentions the structural differences between the CUL2-LRR58 and CUL5-LRR58 complexes but does not provide sufficient context or interpretation of what these differences might mean functionally. A more detailed comparison, including potential implications for substrate geometry or ubiquitination site selection, is needed. The structural portion of the manuscript as a whole is difficult to follow and would benefit from clearer explanation. This paragraph should be revised for clarity and readability.

We have addressed this by adding a more detailed comparison to the discussion and several comparisons to revised Extended Data Figs. 7 and 9.

7. The authors describe extra density between LRR9, the C cap, and CDO1 as “resembling a molecular glue.” This is a somewhat curious analogy, given that the authors then proceed to

model the final four LRRC58 residues into this region based on AlphaFold predictions. The rationale for this interpretation should be clarified. In Extended Data Figure 9, these four LRRC58 residues are modelled into the additional density; however, the quality of the density and the extent to which the model is supported by the map are difficult to assess. In addition, the authors do not discuss any potential interactions between these residues and CDO1, which is surprising given the proposed functional relevance of this motif. To strengthen this interpretation, the authors should report Q-scores for these residues, provide a close-up view of the corresponding map region and identify any putative contacts with CDO1 that could explain why the C-terminal tail of LRRC58 is ordered. Given the apparent functional importance of this motif, mutagenesis of these LRRC58 residues, especially in light of the recent bioRxiv study by Ramage et al., 2025 highlighting the motif as essential for cysteine-mediated instability, would provide a direct test of their contribution to CDO1 binding.

We tried to experimentally address this in two ways. Initially, we hoped to test effects of C-terminal truncations using in vitro ubiquitylation assays. However, despite many attempts, we were unable to purify mutant versions of LRRC58. It is of note that preps of LRRC58 are very low yield, and required large volumes of insect cell culture. Therefore, we tested effects of the mutations by exogenously expressing FLAG-LRRC58 (and mutants) in our newly established LRRC58 knockout cell line, and examined CDO1 levels. Since the CDO1 levels were equally impacted by WT LRRC58 and the deletion mutants, we surmise that the C-terminal residues are not essential for the E3 ligase recognition of CDO1. This was in contrast to the control A266F mutant, designed to disrupt EloB/C binding. We provide these data for the reviewer below.

Based on these data, and the Reviewer's suggestion, we removed the modeled residues from the structure, and replaced the previous text. We now note this region as unassigned density (Extended Data Figure 8b) and speculate that whatever this may be, it might play a role in LRRC58 recruitment of CDO1. We provide an updated ChimeraX file with the corrected PDB file for the reviewer.

[editorial note: figure removed as now publicly available]

Minor points

8. Figure 1d and associated text (lines 138-189). The labelling in Figure 1d is unclear. Improve colour use and match the key order to the bar graph. The statement that “LRR58 amounts change to a striking extent” is not adequately supported by the data as presented. The authors do not quantify fold-changes or statistical confidence, or comment on baseline LRR58 expression differences across cell types.

Thank you for these suggestions. Reevaluating the original figure 1d, we realized that what cell line corresponded to each bar would indeed benefit from clarification. We addressed this (and a related comment from Reviewer #1) in the revised manuscript by labeling each bar with the cell line used. We have toned down our language to not include striking. Unfortunately, we are not able to describe fold-changes as LRR58 was not detectable in most unstarved cell samples we analyzed. We have made changes to the text to reflect this both in the figure and its caption.

9. Figure 5. The colour schemes for RBX1/2 and ARIH1/2 are too similar, making the models difficult to distinguish. More distinct colours are recommended. Panels 5b, 5c, 5d, and 5e are mislabelled in the legend and should be corrected. In Figure 5e, the zinc ion is not visually prominent; a more contrasting colour would improve clarity.

We apologize for lack of clarity and have adjusted our color scheme for better distinction of ARIH proteins from the RBX proteins. We have updated and corrected figure 5b-e legends. We also now show the zinc ion with increased size and in charcoal gray to contrast from the light purple of LRR58.

10. Lines 303-304: The description of the “C cap” and “several LRRs” forming the substrate binding domain is too vague. The authors should specify which LRRs contribute to substrate engagement.

We have clarified which LRRs are involved in substrate engagement.

11. Lines 323-325: The authors state that the LRR58 C terminus could not be fully resolved. Please clarify which residues lack density, and it would be helpful if the disordered regions of LRR58 were highlighted in the LRR58 schematic of Figure 5a.

As mentioned in the response to point 7, we have revised the C-terminal part of the LRR58 structure. We have adjusted the schematic in Figure 5a accordingly.

12. Figure 6a-c. Please label the proteins in these panels, as is done in Figure 5.

Thanks. We have fixed this accordingly.

13. Line 333. The sentence should read: “The ‘top side’ of CDO1 engages a continuous surface with LRR58...” rather than “in LRR58.”

We have fixed this accordingly.

14. Line 245. The authors state that “both cellular and in vitro ubiquitylation assays” were performed, but only a degradation reporter assay is shown for cellular experiments. The authors should clarify this.

Thanks. We have clarified this.

15. Figure 4a. The presentation is confusing for the reader as one is comparing +/- cysteine and the other is +/- compound. Clearer labelling would help.

We have clarified this in the updated Figure 4a.

16. Extended Figure 5 and 6 would benefit from 2D classes being shown similar to Extended Figure 4. Inclusion of local resolution in each figure would also be useful.

Cryo-EM processing of these datasets did not make use of 2D classification. We describe the workflows in the Methods, and have updated the Extended Data Figures to include local resolution maps.

17. The authors describe the limited resolution of CUL2-LRRC58 as due to preferred orientation. However, in Extended Figure 5 the orientation distribution plot shows moderate preferred orientation. Authors should comment on any heterogeneity and continuous flexibility within the sample that could be contributing to the lower resolution.

To quantify the orientation bias and provide insights into the different map qualities, we include cFAR scores in our processing workflows.

Reviewer #3 (Remarks to the Author):

This study investigates LRRC58 as a cysteine-responsive BC-box substrate receptor that targets the metabolic enzyme CDO1 for ubiquitin-mediated degradation. Using a combination of DIA-MS, reporter assays, biochemical reconstitution, and high-resolution cryo-EM of trapped ubiquitylation intermediates, the authors uncover how LRRC58 selectively positions CDO1 for ubiquitylation at Lys8. The structures reveal a previously undescribed LRRC58 architecture with a C-terminal cap domain that engages multiple interaction patches on CDO1. Comparative analyses of CUL2- versus CUL5-assembled CRLs highlight conserved catalytic organization and provide mechanistic insight into selective substrate targeting. Overall, this is a comprehensive and technically impressive study.

We are pleased by the Reviewer’s enthusiasm for our work.

However, several conceptual and interpretational issues, particularly in the early Results sections, limit the strength of some conclusions, where correlation is occasionally interpreted as functional activation.

Major concerns

1. Throughout the manuscript, the authors state that the LRR58–CRL is “activated” upon cysteine starvation. While the authors convincingly demonstrate CRL dependence using MLN4924 and show increased association of LRR58 with neddylated cullins and elevated LRR58 protein abundance under cysteine-starved conditions, these data establish requirement rather than increased enzymatic activity. Enrichment in an activity-based CRL pulldown does not distinguish between increased protein abundance, altered binding stability, reduced deneddylation, or true catalytic activation. Although Figure 3 convincingly demonstrates LRR58-dependent ubiquitination of CDO1 and CRL dependence, these experiments establish biochemical capability rather than cysteine-dependent activation of the LRR58–CRL. Direct measurements of increased ubiquitination activity under cysteine starvation are not provided, and the language should therefore be tempered accordingly.

Thank you for the suggestion. We have reworded the text accordingly.

2. The authors provide convincing genetic evidence that CUL2 is the dominant cullin mediating CDO1 degradation in cells, with CUL5 contributing redundantly, and they further demonstrate that LRR58 can assemble functional CRLs with both cullins in vitro. However, the active-CRL Fab used for profiling recognizes neddylated CUL1, CUL2, CUL3, and CUL4, but not CUL5, meaning that LRR58–CUL5 assemblies are not detectable in the initial discovery experiments. This limitation is not explicitly discussed in the Results section and creates a narrative gap between the early profiling data and the later structural emphasis on CUL5-based complexes. Clarifying whether CUL5 engagement is inferred, context-dependent, or primarily supported by biochemical and structural feasibility would strengthen the coherence of the study.

We apologize for the lack of narrative clarity in our initial manuscript. To address this point, and related comments from Reviewer #2, we now further explain that our activity-based probe only recognizes CUL1, CUL2, CUL3 and CUL4A/B. Because those experiments do not assay CUL5, which was observed several years ago to bind LRR58, we tested both CRLs for ubiquitylation activity and found they are both functional. Thus, we sought cryo-EM data for both CUL2 and CUL5, although the map with CUL5 is higher resolution as the sample with CUL2 had greater orientation bias.

We performed several additional experiments to address reviewer points. First, we examined effects of CUL2 and CUL5 knockdown on endogenous CDO1 in HEK293T cells. Endogenous CDO1 levels increased in low cysteine only upon simultaneous knockdown of both cullins (Figure 3d). Thus, at least in some settings, the two cullins appear functionally redundant. Second, we performed ubiquitylation assays for the CDO1 mutants with CUL5 as well as CUL2. There were no obvious cullin-specific differences. The new data are shown in Extended Data Figures 1a, 4b, 5c-d, 8c of the revised manuscript. We also added brief discussion of these findings and work of others on dual CUL2/5 use of LRR58 and speculate that the different cullins may function in distinct physiological settings.

We also clarify that the knockdown of CUL2, but not of CUL5, was sufficient to stabilize our CDO1 reporter in HEK293T cells. This finding allows extending comparison to the regulation driven by the distinct Compound 8-VHL substrate receptor from the in vitro assays to the reporter stability assays.

3. The authors present convincing evidence that LRR58 is post-transcriptionally regulated in a proteasome- and CRL-dependent manner, as shown by stabilization of LRR58 protein upon MG132 and MLN4924 treatment together with unchanged LRR58 mRNA levels. However, some of the phrasing suggests continuous or constitutive degradation of LRR58 under cysteine-replete conditions, which is not directly measured. Slightly tempering this language would better align the conclusions with the data presented.

We have tempered this wording accordingly. We also added assays showing LRR58 autoubiquitylation in vitro.

4. The manuscript presents strong convergent evidence linking LRR58 to CDO1 degradation through proteomics, genetic dependency, cell-based reporters, biochemical reconstitution, and parallel literature. However, no endogenous LRR58–CDO1 interaction is demonstrated in cells, and all physical interaction evidence derives from in vitro assays or computational predictions. While this may reflect technical limitations, clarifying the predictive nature of these interactions or explicitly acknowledging this limitation would strengthen the in vivo interpretation.

We have now included LRR58 knockout cell data that demonstrates the LRR58 dependent degradation of CDO1 on an endogenous level (Fig. 2g-h).

5. The authors convincingly demonstrate that LRR58-mediated degradation of CDO1 is highly selective for Lys8, whereas compound-8-mediated degradation is lysine-promiscuous. However, the manuscript further concludes that the reduced efficiency of the native pathway relative to degrader-induced degradation primarily results from this lysine selectivity. While this is a plausible explanation, alternative factors such as differences in E2 recruitment, catalytic geometry, initiation versus elongation kinetics, or chain architecture are not examined. Thus, lysine selectivity is clearly supported, but its role as the primary determinant of degradation efficiency is inferred rather than directly demonstrated.

We appreciate the reviewer’s point and additional comments on this topic by Reviewer #2. We have updated the text as follows: “Both cellular degradation reporter assay and in vitro ubiquitylation assays (Fig. 4, Extended Data Fig. 4a) showed that CDO1 targeting was more efficient by targeted protein degradation than through LRR58. This could result from the relatively higher degrader-induced E3-substrate affinity^{54,55}, as well as variation in catalytic geometries that could also impact efficiency of ubiquitylation. To experimentally unveil potential such differences^{44-47,65,66}, we asked if the native degradation mechanism is relatively more constrained by selectivity of lysine targeting. We tested effects of arginine replacements (which cannot accept ubiquitins) for individual lysines in the CDO1 stability reporter (Fig. 4a). Strikingly, a single K8R substitution was impaired for cysteine-dependent destabilization of CDO1. However, all CDO1 variants were readily destabilized by compound-8.”

6. Several structural interpretations, particularly for the CUL2-based complex, appear to extend beyond what is directly supported by map quality. It would be helpful if the authors more explicitly distinguished regions supported by cryo-EM density from those incorporated

via rigid-body docking or AlphaFold predictions. Given that the positioning of Lys8 within approximately 25 Å of the ARIH active site is central to the proposed mechanism, additional information on local resolution and density quality in this region would strengthen confidence in this conclusion. Furthermore, although in vivo data support CUL2 as the dominant cullin for CDO1 turnover, the functional relevance of structural differences between CUL2- and CUL5-based complexes remains unclear and would benefit from further discussion.

We addressed these issues through several changes to the revised manuscript. First, we now show functional redundancy of CUL2 and CUL5 in cysteine-dependent regulation of endogenous CDO1 levels in HEK293T cells. Second, we perform more extensive side-by-side ubiquitylation assays with both CUL2 and CUL5 for mutant versions of CDO1, showing no obvious cullin-specific differences.

We further address this by changes to Figure 5, to better distinguish between the CUL2 fitted model and the experimental CUL5 structure. We now clarify in the text that the CUL2 model was built via docking of subcomplexes, and we explicitly state in the Methods which residues from which structures were used to derive the CUL2 model. We now include local resolution in the updated processing schemes, have added a more detailed comparison of the CUL2 and CUL5 structures to the discussion and several comparisons to a new Extended Data Fig. 9. This new Extended Data figure better shows how LRR58-CUL2 and LRR58-CUL5 place CDO1 adjacent to the active sites of ARIH1 and ARIH2, respectively.

7. Finally, given the existence of a parallel study independently identifying the LRR58–CDO1 pathway, the authors should ensure that claims of novelty throughout the Introduction and Discussion clearly emphasize the unique mechanistic and structural insights provided here, rather than the initial discovery of the pathway itself. In particular, Ramage et al. report LRR58-dependent regulation of CDO1 stability and cysteine homeostasis in vivo, including metabolic and physiological consequences, which places the present work in the context of a parallel pathway discovery and highlights its primary strength in structural and mechanistic resolution rather than pathway identification.

We would like to point out that we extensively cited both the published study (Xiao et. al. Nature) and the preprint (Ramage et al., bioRxiv) in the Introduction, Results, Discussion of our original manuscript, as these studies appeared online while our manuscript was in preparation (two months before our preprint and submission). We now have revised the first sentences of our Discussion section to add yet additional citations to the previous work, and have further edited the Discussion to clarify that our study uniquely:

1. Uses a distinct strategy for discovering a metabolically regulated E3 ligase and substrate using proteomics-driven abundance profiling and structural modeling.
2. Biochemically reconstitutes CDO1 ubiquitylation by both LRR58-CUL2 and LRR58-CUL5 CRL E3 ligase complexes.
3. Compares endogenous and targeted protein degradation mechanisms for CDO1. Our data reveal important mechanistic distinctions.

4. Shows that disease-associated CDO1 mutation sites resistant to LRR58 regulation remain susceptible to targeted protein degradation.

5. Identifies the key lysine required for cysteine-regulated CDO1 turnover in cells, and for LRR58-dependent ubiquitylation in vitro.

6. Resolves cryo-EM maps capturing LRR58-mediated ubiquitylation of CDO1 by both CUL2 and CUL5. These structural data reveal for the first time how LRR58 directs CDO1 for ubiquitylation, how the E3 geometries establish specificity for CDO1 Lys8 targeting, how a single substrate receptor can achieve ubiquitylation with two different cullins, and how a CUL5-RBX2-ARIH2 assembly ubiquitylates a substrate.

Minor concerns

1. Proteomics figures should more clearly distinguish true absence from below-detection abundance.

We have updated the figures and their captions to acknowledge this accordingly.

2. Local resolution overlays at the LRR58–CDO1 interface would improve interpretability.

We now display local resolution side-by-side along the LRR58-CDO1 interface in Extended Data Fig. 6. We now include local resolution maps in the updated processing schemes.

3. Composite figures would benefit from clearer labeling of experimental density versus fitted models.

Figure 5 has been updated to reflect this.

4. Terminology for LRR58 C-terminal regions, for example “C-cap”, should be standardized.

We apologize for the lack of clarity and have updated this where appropriate. We also now make note in the text which residues specifically make up the C-cap region, as also shown in Figure 5.

5. A schematic defining CDO1 interface patches and the D-patch would aid clarity.

Figure 6 now includes a linear schematic defining the interface patches.

6. Brief discussion of potential cooperativity among interface patches would be useful.

We have adjusted the text to include this.

7. Additional contextualization of ARIH1 and ARIH2 architecture relative to prior work would help orient non-specialist readers.

Thank you for the suggestion. Our revision now includes assays showing activity with ARIH1 and ARIH2 relative to other CRL partner ubiquitylation enzymes. Additionally, further

structural views and comparison of ARIH1 and ARIH2 are included in our discussion of the structure and in the revised Extended Data Fig. 7 and 9.

8. In Figure 4b, the time axis annotation in minutes is missing at the indicated position and should be added for consistency with the right panel.

We apologize for the oversight and have corrected this.

Reviewer #4 (Remarks to the Author):

We thank the Reviewer for taking the time and effort to review our study.

RESPONSES TO REVIEWERS' COMMENTS

Reviewer #2 (Remarks to the Author):

the authors have addressed all concerns sufficiently well and clarified all points raised. Happy to recommend it for publication.

We are very pleased that the Reviewer recommends publication of our study in Nature Communications.

Reviewer #3 (Remarks to the Author):

The authors have carefully revised the manuscript and addressed the concerns raised in the initial review. The revised version includes additional experiments, particularly the LRRC58 knockout analyses and expanded biochemical assays, which strengthen the mechanistic link between LRRC58 and CDO1 regulation. The authors have also clarified the interpretation of CUL2 versus CUL5 usage, improved the presentation and explanation of the structural data, and appropriately moderated statements that previously implied catalytic activation of the LRRC58–CRL complex under cysteine starvation.

The structural analysis is now presented more clearly, with improved distinction between cryo-EM density supported regions and modeled components, and the inclusion of local resolution information strengthens confidence in the structural interpretations. The manuscript also better contextualizes its findings relative to recent studies describing the LRRC58–CDO1 pathway.

Overall, the revisions satisfactorily address the major conceptual and technical concerns raised in the previous review. I have no further major concerns and believe the manuscript is suitable for publication.

We are very pleased that the Reviewer recommends publication of our study in Nature Communications.

Reviewer #4 (Remarks to the Author):

We thank the Reviewer for contributing to review of our study..